# A reconciled solution of Meltwater Pulse 1A sources using sea-level fingerprinting

Yucheng Lin [1,2 ✉], Fiona D. Hibbert [2,4], Pippa L. Whitehouse [1], Sarah A. Woodroffe[1], Anthony Purcell [2], Ian Shennan [1] & Sarah L. Bradley [3]

The most rapid global sea-level rise event of the last deglaciation, Meltwater Pulse 1A (MWP-1A), occurred ~14,650 years ago. Considerable uncertainty regarding the sources of meltwater limits understanding of the relationship between MWP-1A and the concurrent fast-changing climate. Here we present a data-driven inversion approach, using a glacio-isostatic adjustment model to invert for the sources of MWP-1A via sea-level constraints from six geographically distributed sites. The results suggest contributions from Antarctica, 1.3 m (0–5.9 m; 95% probability), Scandinavia, 4.6 m (3.2–6.4 m) and North America, 12.0 m (5.6–15.4 m), giving a global mean sea-level rise of 17.9 m (15.7–20.2 m) in 500 years. Only a North American dominant scenario successfully predicts the observed sea-level change across our six sites and an Antarctic dominant scenario is firmly refuted by Scottish isolation basin records. Our sea-level based results therefore reconcile with field-based ice-sheet reconstructions.

[1] Department of Geography, Durham University, Durham, UK. [2] Research School of Earth Sciences, Australian National University, ACT, Canberra, Australia. [3] Department of Geography, The University of Sheffield, Sheffield, UK. [4] Present address: Department of Environment and Geography, University of York, York, UK. ✉email: yucheng.lin@durham.ac.uk

Meltwater Pulse 1A (MWP-1A) was the largest and most rapid global sea-level rise event of the last deglaciation, characterised by ~20 m global mean sea-level (GMSL) rise within 500 years[1,2]. It was driven by the collapse of vulnerable ice sheet sectors, and was concurrent with rapid Northern Hemispheric warming and disruptions in oceanic and atmospheric circulation[3,4]. The ice-ocean-climate feedbacks operating during this period are not well understood largely due to a lack of consensus on the sources of MWP-1A[5,6], which, in turn, were likely to be a key driver in stimulating rapid deglacial climate change[7–9].

Two major techniques have been used to constrain the sources of MWP-1A. The first uses physics-based models, constrained by field-based glacio-geological evidence, to simulate regional ice sheet change during the last deglaciation[10–12]. This approach is restricted by large uncertainties regarding palaeo ice-sheet boundary conditions, climatic conditions, and ice-sheet model parameters. Ice histories derived using this approach do not always match sea-level observations[11]. Conversely, the second method seeks to reconcile ice-sheet change with spatially variable records of sea-level change using a glacio-isostatic adjustment (GIA) model[1,13–15], an approach often termed sea-level fingerprinting[16,17]. The primary limitation of sea-level fingerprinting is the spatial and temporal scarcity of sea-level records across MWP-1A. Commonly, only three sites are used (Tahiti, Barbados and Sunda Shelf), resulting in an under-constrained problem and strongly non-unique solutions[15,17,18]. Other techniques, e.g., analysis of the oceanographic[8,19,20] or isotopic[21] effects of specific meltwater sources, add further constraints, but the primary source of MWP-1A remains controversial with three ice sheets proposed as the major contributor, namely, the North American Ice Sheet, including Greenland (NAIS)[2,10,22], the Antarctic Ice Sheet (AIS)[1,13] and the Scandinavian and the Barents Sea Ice Sheet (henceforth denoted together as SIS)[23].

In this work, we combine a data-driven inversion approach with sea-level fingerprinting to simultaneously determine probability distributions for the magnitude and sources of MWP-1A based on six sea-level sites spanning the far, intermediate and near field. The results indicate a 17.9 m (15.7–20.2 m; 95% probability) global mean sea-level rise during MWP-1A, which consists of a dominant NAIS contribution (accounting for 35–85% of total MWP-1A magnitude), a substantial SIS contribution (20–35%) and a minor AIS contribution (0–35%, with a strong preference for a <15% contribution). Unlike previous sea-level fingerprinting studies[1,13], our results show good agreement with most recent regional ice-sheet reconstructions[10,22–26], and may lead to a reconciled solution of MWP-1A sources.

## Results

**Sea-level fingerprinting approach.** To robustly fingerprint the sources of MWP-1A, three main challenges need to be overcome. First, the above-mentioned non-uniqueness problem. Previously, only three sea-level sites showed sufficient temporal resolution for fingerprinting studies across MWP-1A[15,18], namely, coral reef records from Tahiti[1,27] and Barbados[28–30] and sedimentary indicators from Sunda Shelf[31]. The geographical distributions of these sites do not permit the separation of meltwater sources from the AIS and SIS (Fig. 1). Second, the relationship between coral living depth and environmental conditions, as well as the link between reef accretion and sea-level change[32], is not straightforward and may differ between different localities[33–35]. This can add considerable complexity when interpreting coral sea-level indicators. Third, most previous fingerprinting studies assumed a minor SIS contribution to MWP-1A (1–2.5 m)[1,13,15,18]. This is challenged by a recent chronological reinterpretation of the SIS

ice history that proposes the SIS was a major MWP-1A contributor[23]. Such a large SIS contribution has not yet been tested using sea-level fingerprinting.

We address these challenges via three major methodological advances. First, we increase the number of sea-level sites, with data from extensive coral and coralline algae deposits on the Great Barrier Reef (GBR) at Hydrographer's Passage (HYD) and Noggin Pass (NOG)[36,37], and isolation basin stratigraphies from Northwest Scotland[38–40] (Fig. 1). Where necessary, a standardisation is applied to ensure that the sea-level index points (SLIPs) only reflect the sea-level fingerprint of MWP-1A (see Methods). Second, based on these standardised SLIPs, we estimate the local magnitude of relative sea-level (RSL) change across MWP-1A at each of our six sites using a Monte Carlo (MC) linear regression approach to capture the vertical and chronological uncertainties of the sea-level indicators. Third, these local MWP-1A magnitudes of RSL change are used to invert for the global magnitude, and regional partitioning of meltwater via fingerprinting of NAIS, AIS and SIS change.

Our approach relies on the assumption that SLIPs can be used to map out the fingerprint of ice-sheet change. A sea-level fingerprint reflects the global geoid variation and instantaneous elastic solid Earth response to ice mass change[16,17], also known as the elastic component of RSL change[18]. Assuming the NAIS, AIS and SIS were the only contributors to MWP-1A, the global pattern of RSL change caused by melt from these ice sheets can be identified as a linear combination of three spatially variable sea-level fingerprints (Fig. 1d–f), each scaled by the eustatic contribution from the related ice sheet[17]. Before using SLIPs to map out the fingerprint of MWP-1A, three issues must be addressed (i.e., our standardisation). The first only affects our Northwest Scotland data. RSL change here contains a large local GIA signal associated with changes to the British-Irish Ice Sheet (BIIS). We determine a local GIA correction for all these SLIPs to isolate the GIA signal associated with non-local ice sheet change[14,41] (see Methods, Fig. 2f). The second issue concerns the spatial gradient of the sea-level fingerprint between coring locations within one site (e.g., >10km wide)[15]. This affects the Sunda Shelf and Northwest Scotland data and is accounted for by subtracting the time-specific sea-level gradient between each SLIP and a reference location. The corrected SLIPs represent RSL at a single locality (red stars in Fig. 2c,f and Supplementary Fig. 2, see Methods). Lastly, for all SLIPs, we remove the age- and location-specific viscous component of RSL change. This correction accounts for the viscous solid Earth response to changes in surface loading and the accompanying change in geoid height caused by ice-sheet variation prior to and during MWP-1A (see Methods). We determine all three corrections using the mean of a GIA model ensemble that accounts for uncertainties associated with global ice history and mantle properties (see Methods). Because we focus on the centennial timescale of MWP-1A, our GIA corrections are not strongly sensitive to choice of mantle properties (see Supplementary Fig. 3), and hence neglect of heterogeneity (i.e., 3-D Earth structure)[42] should not bias our results.

**Estimating local sea-level change across MWP-1A.** The standardised SLIPs constrain local MWP-1A magnitude at each site. We use a conservative time window of 14.65-14.00 ka BP to select SLIPs at each site that clearly mark the initiation and termination of MWP-1A, enabling us to capture the full magnitude of MWP-1A sea-level rise. Ideally, only records with mean ages within this window will be selected, but for sites with insufficient temporal coverage we include records whose 2σ age error bars extend into it. We estimate local MWP-1A magnitude from the SLIPs using a

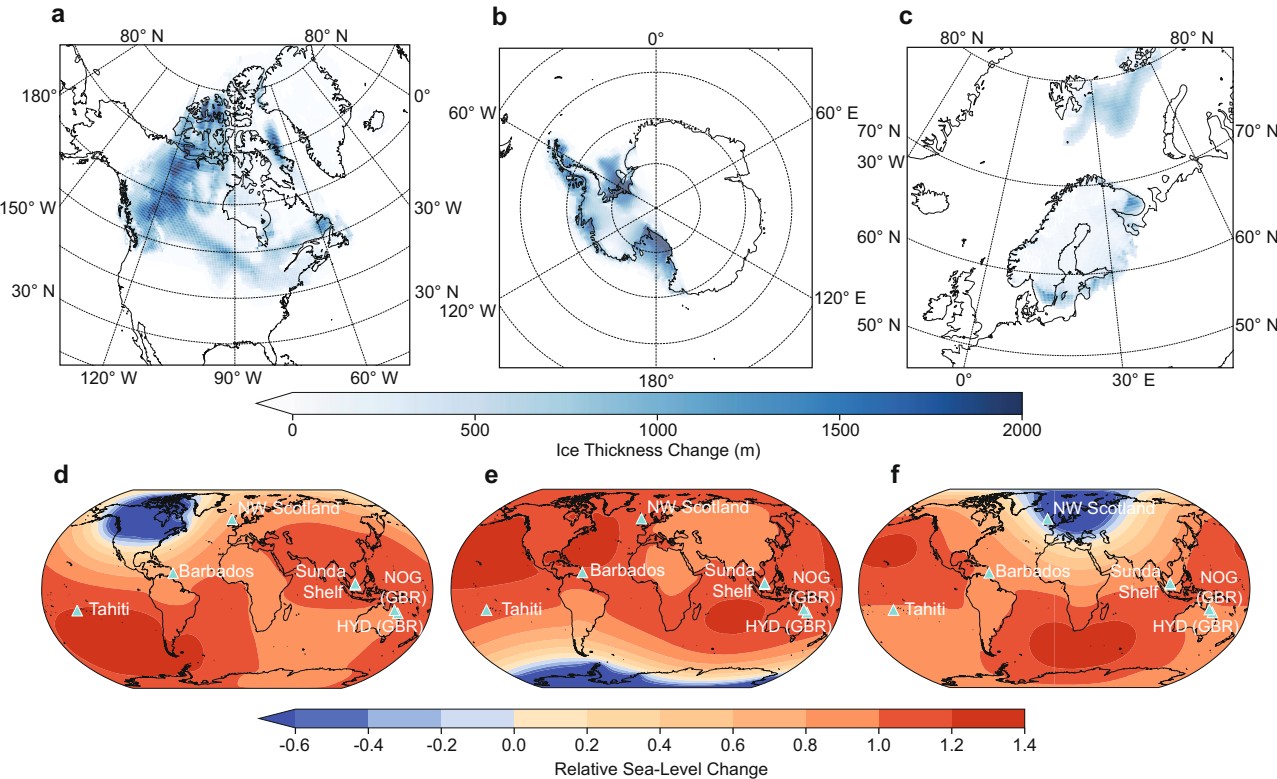

**Fig. 1 Ice melt geometries and normalised sea-level fingerprints used in this study. a–c** Ice mass loss pattern for the NAIS, AIS and SIS, which were used to generate sea-level fingerprints shown in (**d–f**) that represent elastic-induced global relative sea-level change corresponding to one unit of ice mass loss from each ice sheet. **a, c** The reconstructed MWP-1A ice melt geometries from Lambeck et al.[24] and BRITICE-CHRONO[52]. **b** The LGM-to-present West AIS melt geometry from Whitehouse et al.[54] (see Methods for details). The cyan triangles and text denote the location and name of each sea-level site. HYD = Hydrographer's Passage, NOG = Noggin Pass, GBR = Great Barrier Reef, NW Scotland = Northwest Scotland.

MC linear regression method, which captures any asymmetric depth and age uncertainties of different types of sea-level indicators by randomly sampling their uncertainty distributions. We use two approaches to represent indicative depth distributions of coral sea-level indicators; an empirically-derived distribution from modern coral analogues (the empirical scenario) and a uniform distribution, using palaeo-water depths from original publications (the uniform scenario)[35,43]. For non-coral SLIPs, both scenarios adopt a uniform indicative depth distribution based on original publications. The MC sampling process also accounts for the error propagation associated with any GIA correction applied and elevation measurement uncertainties (see Methods). We calculate chronological probability distributions following the methodology of Hibbert et al.[43], accounting for multimodal, asymmetric [14]C age distributions and age reliability screening (see Methods).

MC linear regression, using randomly selected data points and a weighted least square method, determines the local MWP–1A RSL rise rate at each site (see Methods). We convert this to local MWP-1A magnitude by multiplying by the assumed duration of MWP-1A (500 years in this study). We exclude, as implausible, regressions producing a reverse slope (i.e., a sea-level fall). Repeating this process 20,000 times (excluding the implausible iterations) produces distributions of local MWP-1A magnitude for each site (Figs. 2, 3a). Because our results are derived from the averaged sea-level rise rate throughout MWP-1A, they are linearly scalable to any assumed duration of MWP-1A.

The viscous component of RSL change has a significant effect on local sea-level change during MWP-1A. Far-field localities (Tahiti, Sunda Shelf, HYD and NOG) will have experienced local sea-level fall associated with the redistribution of water to regions experiencing peripheral bulge subsidence and due to the ocean load-induced continental levering effect (we refer to the combined effect as ocean siphoning[44]; Supplementary Fig. 3 and S4). Not considering this effect would lead to ~1 m underestimate of the local RSL magnitude. Conversely, Northwest Scotland will have experienced 0.8 m local RSL rise during MWP-1A due to subsidence of the SIS peripheral bulge. Being an intermediate-field site, Barbados experienced both ocean siphoning and peripheral bulge subsidence during MWP-1A. The effects of these two signals roughly balance each other (Supplementary Fig. 3 and S4). It should be noted that given the exponential-like form of postglacial decay, the non-linear viscous signal associated with ice melt during MWP-1A is approximately double the linear pre-MWP-1A viscous signal (see Methods), a point largely unconsidered in previous work[15]. We recommend both viscous signals be considered in future meltwater source inversion studies.

At Tahiti, our inversion is tightly constrained by samples containing vermetid gastropods (yellow error bars in Fig. 2a) that indicate very shallow environments (<5–6 m[1,45,46]). Most of the other Tahiti coral samples were identified as *Porites* sp.[1]. Modern analogues (the empirical scenario) suggest a bimodal depth-habitat distribution concentrated at 0–15 m and 40–50 m[43]. This bimodal empirical distribution was generated from a global compilation, and given insufficient modern observations at Tahiti, we consider it a maximum vertical depth range for this species. Comparatively, the palaeo-water depths derived from coral-algal assemblages (our uniform scenario) suggest depths of 0–10 m or even 0–20 m[46,47]. Therefore our empirical scenario yields a larger uncertainty range for the MWP-1A magnitude (13.6–30.9 m for a 500-year duration; 95% confidence interval;

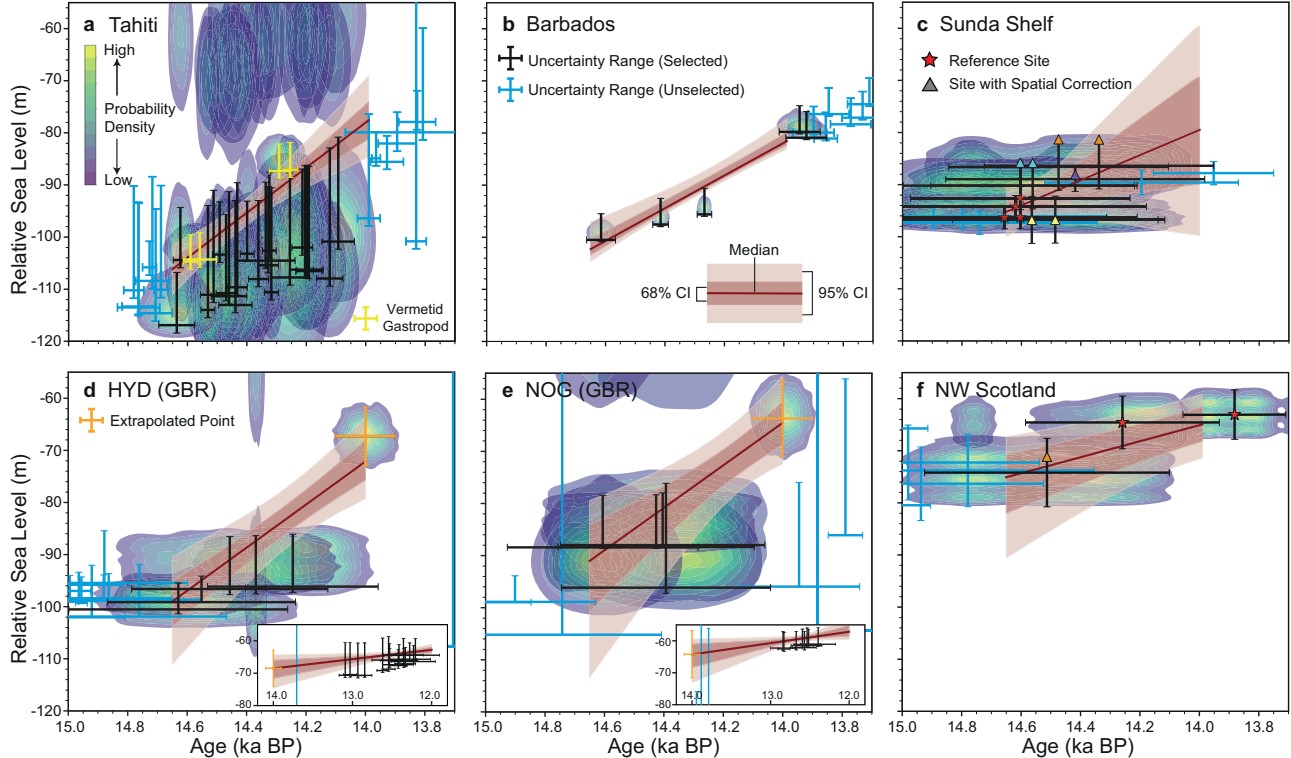

**Fig. 2 Estimated local MWP-1A sea-level rise trend at six selected sites using our empirical scenario.** Blue-green-yellow shading indicates the data-point-specific probability density accounting for age and depth uncertainties. The depth uncertainty for coral-based SLIPs (**a**, **b**, **d**, **e**) was determined using modern ecological data (the empirical scenario), some of which present a bimodal habitat depth, resulting in complex data clouds. For the uniform scenario, see Supplementary Fig. 6. The median and 95/68% confidence interval (CI) were determined by 20,000 MC simulations (see main text). Black/blue error bars reflect uncertainty ranges associated with viscous and spatial signal corrections applied to SLIPs that were selected/unselected to train the MC model. Vertical and horizontal bars indicate combined depth uncertainty and 2σ age error. **a** The selected coral samples containing vermetid gastropods are highlighted in yellow. **c**, **f** A spatial sea-level gradient correction was applied to ensure all SLIPs represent sea level at a single locality (reference site, denoted by red stars) in Sunda Shelf and Northwest Scotland. Triangles with different colours correspond to different localities, their vertical position indicates the original elevation prior to spatial gradient correction. **d**, **e** Orange error bars indicate the extrapolated point, the extrapolation process is shown in the subplot. **f** SLIPs have been corrected for the local GIA signal using the BRITICE-CHRONO ice model with 120 Earth models. GIA modelling uncertainty is incorporated into the error bars and data clouds.

CI) than the uniform scenario (15.5–26.6 m, see uniform scenario results in Supplementary Fig. 6). Under a 340-year duration, our result suggests a 14.5 m sea-level rise (18.7 m if only using the vermetid gastropods records), similar to a previous estimate of 12–22 m[1].

Because the two Great Barrier Reef sites experienced reef demise and landward migration across MWP-1A, SLIPs from HYD and NOG only show a rapid ~10 m sea-level rise ~14.6–14.4 ka BP with no clear post-MWP-1A marker until the initiation of new coral reefs at ~13.0 ka BP (Fig. 2d,e)[36,37]. To estimate the MWP-1A magnitude at these sites, we determined RSL at 14.0 ka BP by extrapolating back from the large number of SLIPs between ~13.0 and 12.0 ka BP (see Methods and Supplementary Fig. 5). Based on the extrapolated points, our bimodal empirical distribution yields larger MWP-1A uncertainty ranges of 12.0–32.7 m and 7.3–37.7 m for HYD and NOG (95% CI; 500 years) than the counterpart generated by the uniform scenario (9.3–31.9 m and 11.5–28.2 m, Fig. 3a), with both showing good agreement with Tahiti.

At Sunda Shelf, temporally clustered SLIPs with ~0.4 ka age uncertainties (2σ; Fig. 2c) provide a poor constraint on RSL rise, with ~35% of MC simulations producing a reverse slope. We therefore only use weighted least square (without MC simulation) to calculate the local MWP-1A RSL magnitude, fitting to the mean of the age/depth distribution of each SLIP, which was

assumed to be normally distributed. We exclude one data point, from site 18302 (blue error bars in Fig. 2c), because it is inconsistent with other SLIPs from this region (dated ~14.2 ka BP but suggests 5 m lower RSL than SLIPs at 14.4 ka BP) and would strongly bias the local MWP-1A magnitude estimation. These modified regression conditions, combined with the 2-5 m between-site sea-level gradient corrections (Supplementary Fig. 2), produce a large uncertainty range for the MWP-1A magnitude, 0–35.7 m (95% CI, median 15.5 m). Compared with MWP-1A magnitude estimates for other far-field sites, this median value is slightly lower (Fig. 3a), likely due to SLIPs from site 18301 (yellow triangles in Fig. 2c) indicating 10-15 m lower RSL than other SLIPs of a similar age. The RANSAC outlier detection algorithm[48] suggests, with >90% probability, that these index points are outliers, and excluding them yields a ~21.7 m MWP-1A magnitude. However, we choose to retain them for our analysis, and the large uncertainty range, because the MWP-1A partitioning results do not strongly depend on the local MWP-1A magnitude at Sunda Shelf (see Supplementary Note 2).

At Barbados, a recent coral-based sea-level reconstruction[30] significantly improved the temporal control on local RSL at the termination of MWP-1A; two samples at ~14.0 ka BP in Fig. 2b were not available to former studies[1,15]. Constrained by these new SLIPs, both empirical and uniform scenarios yield a tight 95% confidence range of 12.1–20.0 m and 12.8–18.2 m with a common

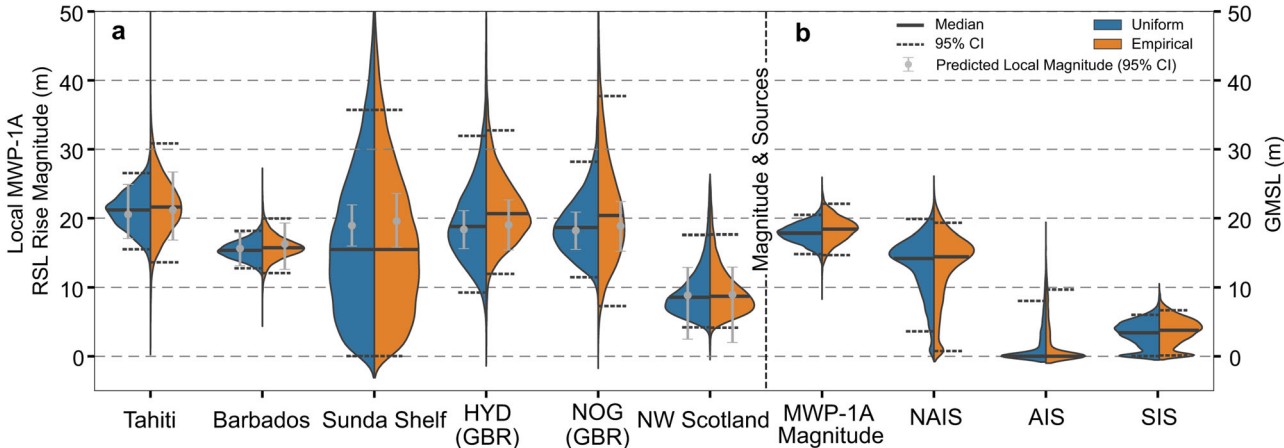

**Fig. 3 Inversion results of local MWP-1A RSL rise magnitude and MWP-1A sources.** Probability distributions of (**a**) local MWP-1A RSL rise magnitudes, (**b**) total MWP-1A magnitude and contribution from each ice sheet assuming a 500-year duration are shown as a violin plot. The two sides of each violin plot correspond to the empirical (orange) and uniform (blue) scenarios used to represent coral living depths. For non-coral SLIPs (Sunda Shelf and Northwest Scotland), both scenarios adopt a uniform distribution, small differences between the two sides are associated with different viscous signal corrections. Grey error bars in (**a**) represent the predicted local MWP-1A magnitude (95% probability) calculated using the inverted MWP-1A sources in (**b**). All probability density functions are derived by Gaussian kernel density estimation, and all inversion results are exclusively non-negative.

median of ~15.5 m (500-year MWP-1A duration; Fig. 3a). Linearly scaling to 340 years yields a median of 10.7 m, which is lower than previous estimates of ~15 m[1] or 9.7–33.6 m[15], and lower than our estimated MWP-1A magnitudes at other far-field sites.

A low MWP-1A magnitude is also observed in Northwest Scotland. After correcting for the local GIA signal and the spatial sea-level gradient, we identify an 8.6 m MWP-1A magnitude (500-year duration) within a 95% confidence range of 3.9–17.3 m (Fig. 3a). The majority of this uncertainty is associated with the three ice models used to determine the local GIA signal, ANU[2,49], PATTON2017[50,51] and BRITICE-CHRONO (S. Bradley, personal communication 2020)[52], with only minor uncertainty associated with Earth parameters (see Methods and Supplementary Fig. 7). Although the three BIIS models yield relatively large differences regarding the magnitude of the local GIA signal, they provide good consensus on the local elastic-induced MWP-1A sea-level rise magnitude: 9.0 m for ANU, 8.9 m for PATTON2017 and 7.7 m for BRITICE-CHRONO (see Supplementary Note 1). The low MWP-1A magnitude observed in Barbados and Northwest Scotland indicates a dominant contribution to MWP-1A from their nearby ice sheets (i.e., the SIS and NAIS, see Fig. 1).

**MWP-1A source inversion.** For each of our 20,000 MC simulations of local MWP-1A magnitude, we adopted a non-negative weighted least square algorithm[53] to optimise the contribution of the NAIS, AIS and SIS to MWP-1A based on sea-level fingerprints generated using realistic deglaciation geometries[24,52,54,55] (Fig. 1a–c; see Methods). We also tested alternative sea-level fingerprints based on MWP-1A ice melt geometries from ICE6G_C[22], GLAC-1D[10] and G12[11,18] for the NAIS and PATTON2017[50,51] for the SIS, which results in a negligible difference to our results (Supplementary Table 3). The optimisation process was repeated six times, each time removing one site from the six-site database to quantify the bias associated with data over-dependency and assess the consistency of the overall results (i.e., jackknife resampling). We achieve a bias-corrected inversion of MWP-1A magnitude and sources by subtracting the bias (i.e., the difference between overall jackknife ensemble mean and original results) from the original inversion result (Fig. 3b). The averaged 95% CI of the empirical and uniform scenarios gives GMSL rise during MWP-1A between 15.6 and 20.3 m (mean

17.9 m, Fig. 3b). The SLIPs prefer a dominant NAIS contribution to MWP-1A of 13.1 m (6.0–18.3 m), a substantial contribution from the SIS of 3.3 m (0.5–6.0 m) and a small contribution from the AIS of 1.5 m (0–6.9 m). The jackknifing results (i.e., inversion results when each site is excluded in turn) are generally in agreement (Supplementary Fig. 8), pointing to a dominant NAIS contribution and a minor AIS contribution, but they highlight the non-uniqueness of the solution when near-field sites are excluded (Supplementary Note 2).

Our GMSL rise magnitude is primarily constrained by data from Tahiti, Sunda Shelf, HYD and NOG because they are relatively insensitive to the origins of the meltwater (Fig. 1), in contrast to Barbados and Northwest Scotland. For Barbados, melt from the NAIS is the only scenario that produces a considerably reduced local sea-level rise (20% less than the global mean, Fig. 1d). A dominant NAIS contribution is therefore required to produce ~15.5 m sea-level rise at Barbados under a 17.9 m GMSL rise scenario. NAIS melting also results in reduced RSL rise in Northwest Scotland (25% less than the global mean), but to match the observed 8.6 m sea-level rise at this site (<50% of the GMSL magnitude) requires a significant MWP-1A contribution from the SIS. Our inversion approach for the partitioning of melt between the NAIS, AIS, and SIS, successfully reproduces the local MWP-1A magnitude at our six sites (Fig. 3a, grey error bars).

Our inversion results are used to predict deglacial RSL change at our six sites by incorporating our MWP-1A ice history into the ANU ice model (denoted the ANU_MWP model, Fig. 4h; see Methods). The RSL predictions (Fig. 4a–f) show good fit to the data at all six sites. In particular, predictions at four sites across Northwest Scotland show monotonic sea-level fall during MWP-1A (Fig. 4g), which is supported by the stratigraphic interpretation of isolation basins that were isolated shortly before or during MWP-1A, and where no RSL oscillation is recorded (Supplementary Note 3)[38–40,56]. This condition of no RSL oscillation during MWP-1A can only be achieved if the rate of RSL rise due to far-field melt did not outpace land uplift due to local GIA (detailed interpretation in Supplementary Note 3). Modelling of the local GIA signal suggests the largest plausible rate of land uplift at Arisaig (one of the sites in Northwest Scotland) is 9.8 m in 500 years (Supplementary Note 3). After accounting for 0.8 m sea-level rise caused by subsidence of the SIS peripheral bulge, to avoid a local sea-level oscillation, the RSL fingerprint of MWP-1A

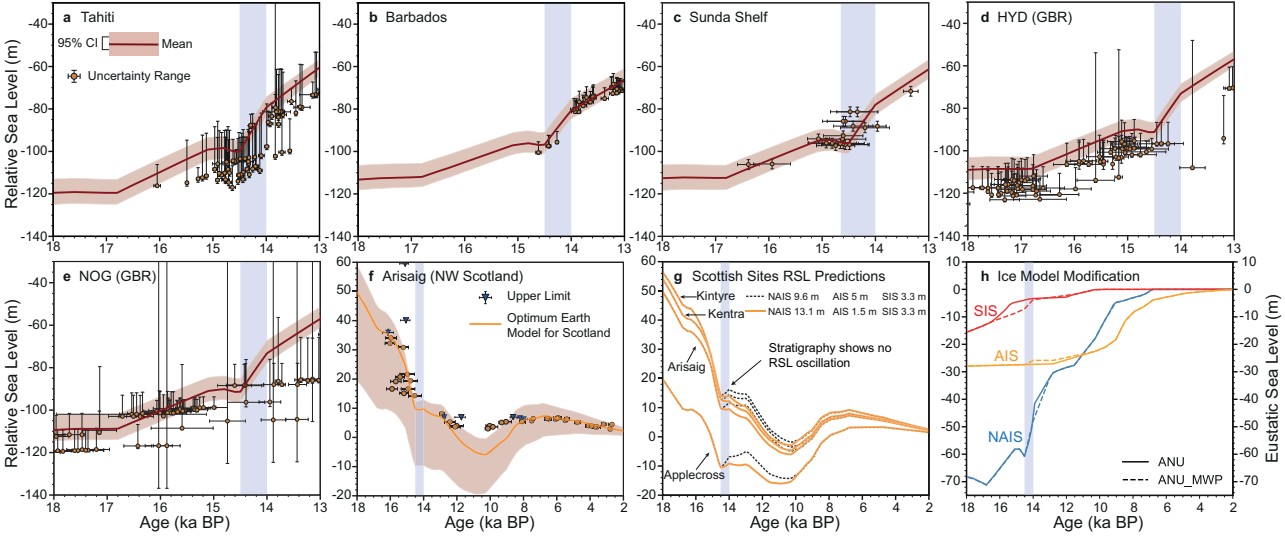

**Fig. 4 RSL predictions using the modified ANU model (ANU_MWP).** Error bars show depth range and 2σ age uncertainties provided by original studies. The blue vertical band indicates the duration of MWP-1A assumed in the ANU ice model (14.5–14.0 ka BP). **f** Orange solid line indicates the RSL prediction generated using the optimum Earth model (65 km lithospheric thickness, 4/200 × 10²⁰ Pa s upper/lower mantle viscosity) instead of the ensemble mean as in (**a**–**e**). By combining our MWP-1A solution with this optimum Earth model in Scotland, we achieve a good fit to RSL data and meanwhile avoid a local RSL oscillation. **g** RSL predictions for four Northwest Scotland sites generated using the optimum Earth model in combination with two MWP-1A scenarios: one that uses the ensemble mean inversion result of this study (orange solid lines) and one where the AIS contributes 5 m to MWP-1A (black dashed lines). The isolation basin stratigraphies indicate no RSL oscillation in Northwest Scotland during MWP-1A. **h** Ice history modifications, solid and dashed lines represent the ANU and ANU_MWP model, respectively. Note the different axis.

cannot exceed ~9 m within 500 years. We refer to this as the sea-level oscillation limit. Under the scenario of 17.9 m GMSL rise, this 9 m limit is exceeded for any substantial AIS contribution because sea-level rise due to melt from the AIS is amplified by 10% across Scotland (Fig. 1e). We assessed the potential consequence of a 5 m AIS contribution to MWP-1A (with the NAIS contribution equivalently reduced, see Methods). This produces a distinct 2,000-year RSL oscillation following the start of MWP-1A (black dashed lines in Fig. 4g). The stratigraphic evidence firmly refutes such an oscillation[56]. In summary, the isolation basin evidence supports a minor AIS, a substantial SIS and a dominant NAIS contribution scenario.

We recalculate uncertainty ranges for our inversion results by imposing a 9 m upper limit on the local MWP-1A magnitude in Northwest Scotland to avoid a local sea-level oscillation. The resulting 95% probability range of the averaged empirical and uniform scenarios suggests a total GMSL rise of 17.9 m (15.7–20.2 m), which consists of a dominant NAIS contribution of 5.6–15.4 m (accounting for 35-85% of total MWP-1A magnitude), a substantial SIS contribution of 3.2–6.4 m (20–35%) and a minor AIS contribution of 0–5.9 m (0–35%) with median values of 12.0, 4.6 and 1.3 m, respectively (Fig. 5).

## Discussion

Our estimates show good agreement with recent field-based ice-sheet reconstructions for the NAIS and AIS (Supplementary Table 2). Conversely, most SIS regional reconstructions propose a 1–2.5 m contribution[2,51,55], considerably lower than our estimate. A possible reason for this discrepancy is previous studies are commonly based on radiocarbon chronology that assumes a temporally constant Scandinavian marine radiocarbon reservoir age, which suggests the southern Barents Sea sector collapsed well before MWP-1A (see Fig. 4h). A recent study adopts a temporally varying Scandinavian marine radiocarbon reservoir age to rein-terpret the chronology of SIS retreat and suggests that the southern Barents Sea sector may have collapsed during MWP-1A,

accompanied by marginal retreat of the Scandinavian Ice Sheet, contributing 4.0–7.4 m to GMSL rise (we calculate the eustatic contribution by subtracting the volume of ice below flotation, as defined in the PATTON2017 ice model[51], from the total ice volume change[23]), similar to our estimate. We suggest a substantial SIS contribution is essential to reconcile the gap between regional ice-sheet reconstructions and global sea-level fingerprinting (see Supplementary Table 2), and thus close the MWP-1A global sea-level budget. Such a substantial freshwater input to the Nordic Sea (~0.12 sverdrup), combined with NAIS freshwater routed along the Mackenzie River into the Arctic, may have contributed to a weakening of this limb of the Atlantic Meridional Overturning Circulation[57], potentially helping to explain the termination of Bølling warming and the initiation of the Older Dryas stadial[8,19,58].

Based on our inversion results, we hypothesise that MWP-1A was triggered by rapid disintegration of Northern Hemispheric ice sheets, which account for at least 65% (95% probability) of GMSL rise during this period. Rapid disintegration of the NAIS and SIS has been proposed to be consistent with the operation of ice-sheet saddle collapse[11,59] and unstable grounding line retreat[60,61] forced by abrupt Northern Hemispheric atmospheric and oceanic warming[3,57]. However, the detailed ice dynamic behaviour of these two ice sheets remains elusive. Although most recent studies suggest that saddle collapse between the Western Laurentide and Cordilleran Ice Sheets was a major contributor to MWP-1A[10,11,18,22,24], a recent study based on the Bering Strait flooding history suggests this saddle collapse did not operate until the Younger Dryas and the NAIS contribution to MWP-1A solely originated from the Eastern Laurentide Ice Sheet[62]. Similarly, for the SIS, the new southern Barents Sea sector collapse chronology proposed by Brendryen et al.[23] is yet to be validated.

To test the sensitivity of our inversion results to alternative ice melt configurations, we separated the NAIS into the Western and Eastern NAIS (separated by 110°W, as defined in Pico et al.[62]) and generated two sea-level fingerprints using the ICE6G_C model[22]. We solve for the contribution of these two NAIS sectors to MWP-1A separately along with the AIS and SIS. The results

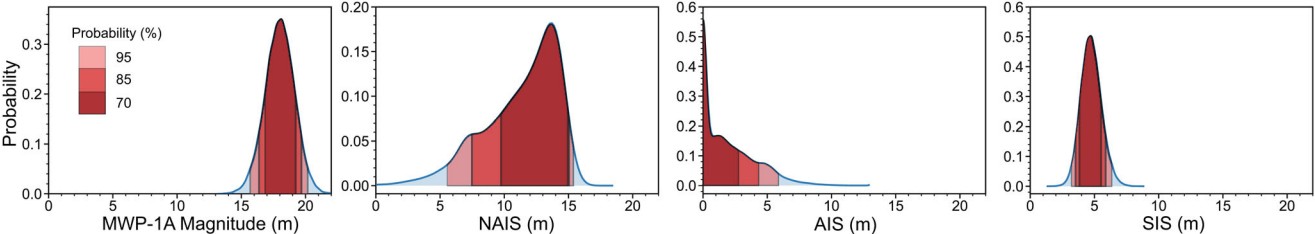

**Fig. 5 Sea-level oscillation limit constrained inversion results.** Each panel shows a probability density function of the averaged inversion result of the empirical and uniform scenarios associated with total MWP-1A magnitude or the contribution from each ice sheet. Shaded areas indicate 95/85/70% probability range.

yield 4.4 m (0–12.5 m; 95% CI) and 8.7 m (0–16.8 m) contributions from the Western and Eastern NAIS, respectively, with little change to the AIS and SIS contributions (Supplementary Table 3). The total NAIS contribution is similar to our original inversion and this NAIS partitioning is consistent with recent NAIS reconstruction studies[10,22,24,63], but due to the limited sea-level sites available, we cannot robustly determine the relative contribution from these two NAIS sectors. For the SIS, we replace the adopted SIS sea-level fingerprint (Fig. 1f, reflecting ice melt from northern Barents Sea and eastern Fennoscandia) with a fingerprint generated using an ice melt geometry that is predominantly sourced from the southern Barents Sea (from the PATTON2017 model[51]). The impact on the inferred MWP-1A contribution from each ice sheet is <0.3 m (Supplementary Table 3). Therefore, our results are not strongly sensitive to the assumed ice melt geometry.

Our results suggest the AIS was relatively stable during the concurrent Antarctic Cold Reversal[64], which is consistent with recent AIS modelling studies[25,26,65]. Based on our melt geometries, MWP-1A induced 15–18 m local RSL rise around Antarctica. This may have started to destabilise the AIS[25,65,66], eventually leading to substantial AIS retreat from 13 ka BP[67].

Our inversion, which includes sophisticated treatment of uncertainties associated with sea-level data and geophysical modelling processes, provides calibrated MWP-1A contributions from the NAIS, AIS and SIS that are consistent with both sea-level constraints and regional ice-sheet reconstructions. In particular, our MWP-1A partitioning is supported by Scottish isolation basin stratigraphies, which can only be fit by a minor Antarctic contribution. Use of our results to prescribe the global pattern of meltwater discharge during MWP-1A[68] may yield novel insights into the sequencing of ice-ocean-climate interactions during this recent abrupt climate change event.

## Methods

**Inversion strategy.** We solve for the meltwater contributions from the NAIS, AIS, and SIS that bestfit observations of RSL change across MWP-1A ($\Delta RSL_{Obs}$). RSL change takes place due to changes in the shape of the solid Earth and the sea surface, with the latter being defined by the shape of Earth's gravity field. Because the solid Earth behaves viscoelastically over the timescale of interest, RSL change at location $\varphi$ can be divided into a component that reflects the instantaneous response of the solid Earth and sea surface to an influx of meltwater ($\Delta RSL_{Elastic}$) and a component that reflects ongoing perturbations to these surfaces due to past surface load change ($\Delta RSL_{Viscous}$):

$$\Delta RSL_{Obs}(\varphi) = \Delta RSL_{Viscous}(\varphi) + \Delta RSL_{Elastic}(\varphi) \quad (1)$$

The elastic term can be further decomposed into:

$$\Delta REL_{Elastic}(\varphi) = ESL_{NAIS} \times F_{NAIS}(\varphi) + ESL_{AIS} \times F_{AIS}(\varphi) + ESL_{SIS} \times F_{SIS}(\varphi) \quad (2)$$

where the three ESL terms represent eustatic sea-level (ESL) contributions from the NAIS, AIS, and SIS ice sheets and the $F(\varphi)$ terms denote ice-sheet-specific, site-specific sea-level fingerprint values. The term sea-level fingerprint describes the normalised elastic component of RSL change triggered by a given pattern of ice loss[16,17]. $F_i(\varphi)$ is insensitive to the value of $ESL_i$ and the choice of Earth rheology[17,69]. Therefore, it can be calculated using existing ice sheet

reconstructions and treated as a known parameter (see Sea-level fingerprint). The three ESL terms are the unknown parameters in our inversion. We assume that the NAIS, AIS, and SIS are the only contributors to MWP-1A, where any Greenland contribution is included in the NAIS.

The viscous component of RSL change can also be decomposed into two terms:

$$\Delta RSL_{Viscous}(\varphi) = \Delta RSL_{PreViscous}(\varphi) + \Delta RSL_{DurViscous}(\varphi) \quad (3)$$

where $\Delta RSL_{PreViscous}(\varphi)$ and $\Delta RSL_{DurViscous}(\varphi)$ are the changes associated with the viscous effects of ice melt prior to and during MWP-1A respectively (see Viscous component of sea-level change).

Substituting Eq. 3 into Eq. 1 yields an expression for the elastic component of RSL change at each field site:

$$\Delta RSL_{Elastic}(\varphi) = \Delta RSL_{Obs}(\varphi) - \Delta RSL_{PreViscous}(\varphi) - \Delta RSL_{DurViscous}(\varphi)$$
$$= ESL_{NAIS} \times F_{NAIS}(\varphi) + ESL_{AIS} \times F_{AIS}(\varphi) \quad (4)$$
$$+ ESL_{SIS} \times F_{SIS}(\varphi)$$

By deriving estimates of $\Delta RSL_{Elastic}$ at our six field sites we create a set of equations that can be inverted to yield the ESL contribution to MWP-1A from each of the three ice sheets considered here (see Inversion for MWP-1A sources).

Monte Carlo linear regression is used to estimate probability distribution of the $\Delta RSL_{Elastic}$ at each site by computing the fit to probability distributions of all viscous-corrected sea-level index points (SLIPs) at that site that lie within MWP-1A (see Monte Carlo linear regression). Prior to carrying out the linear regression, the SLIPs are also corrected for any local GIA effects (see Local GIA signal in Northwest Scotland) and any spatial gradient of RSL that exists between sites that are combined to estimate sea-level change at a single location (see Spatial sea-level gradient). We assume the thermosteric contribution to RSL change during MWP-1A is negligible.

Because $\Delta RSL_{PreViscous}(\varphi)$ is controlled by ice melt prior to MWP-1A it can be calculated using an existing global ice model (see below). In contrast, $\Delta RSL_{DurViscous}(\varphi)$ depends on the unknown ESL parameters, which makes Eq. 4 an implicit equation that must be solved iteratively. We employ the following approach (see Supplementary Fig. 1): (i) Calculate the first approximation of $\Delta RSL_{Elastic}(\varphi)$ at six sea-level sites (see main text) using a Monte Carlo linear regression method that assumes $\Delta RSL_{DurViscous}(\varphi)$ is zero. (ii) Invert for the three ESL values using the $\Delta RSL_{Elastic}(\varphi)$ values calculated in step i (for the first iteration) or step v (for all other iterations). (iii) Correct the bias within the inversion results using jackknife resampling (see details below). (iv) Calculate $\Delta RSL_{DurViscous}(\varphi)$ using the bias-corrected ESL inversion from step iii (see details below). (v) Computing $\Delta RSL_{Elastic}(\varphi)$ using the $\Delta RSL_{DurViscous}(\varphi)$ values obtained in step iv. (vi) Repeat step ii-v until convergence of ESL values has been achieved.

**GIA modelling.** Sea-level change and the solid Earth response to changes in surface loading are computed using the CALSEA software package[70,71], which uses a gravitationally self-consistent theory that accounts for migrating shorelines and Earth rotational feedback[72–75]. The Earth is represented by a spherically symmetric, radially stratified (i.e., 1-D), self-gravitating Maxwell body comprising an elastic lithosphere, and an upper and lower mantle extending to 670 km and from 670 km to the core-mantle boundary, respectively. The elastic and density structure of the Earth is derived from the preliminary reference Earth model[76]. GIA modelling is used to calculate sea-level fingerprints, the local GIA signal in Northwest Scotland, spatial sea-level gradients, and the viscous component of sea-level change.

**Sea-level fingerprint.** The sea-level fingerprint for each ice sheet is obtained by calculating the normalised global pattern of RSL change associated with melt from a specific ice sheet over a finite time interval. Because sea-level fingerprints are sensitive to the geometry of ice melt[69], we use realistic melt geometries across MWP-1A from two recent regional ice-sheet reconstructions of Lambeck et al.[24] and the BRITICE-CHRONO project (with SCEAN1D scenario; S. Bradley, personal communication 2020) for the NAIS and SIS, respectively (Fig. 1a, c). The latter is constrained using geomorphological data compiled in Hughes et al.[55] and Clark et al.[52], and reconstructed using a plastic ice-sheet model[77]. We also used

some alternative NAIS and SIS melt geometries from ICE6G_C[22], GLAC-1D[10] and G12[11] for the NAIS and PATTON2017[50,51] for the SIS to test the dependence of the inversion results on the assumed ice melt geometries. This leads to essentially unchanged inversion results (see Supplementary Table 3). For the AIS, due to the lack of geological constraints, the melt geometry across MWP-1A remains largely unknown. Since East Antarctica is estimated to have contributed only ~1 m to post-Last Glacial Maximum ESL rise, with this melt most likely to have occurred after MWP-1A[78], any Antarctic contribution to MWP-1A is likely to have come from the West AIS. We therefore generate the AIS sea-level fingerprint using the Last Glacial Maximum-to-present pattern of ice loss across West Antarctica adopted by the W12 ice model[54] (Fig. 1b). Because all our six sea-level sites are far away from the AIS, their AIS sea-level fingerprint values are not sensitive to the assumed West AIS melt geometry.

**Local GIA signal in Northwest Scotland.** As demonstrated in previous studies[14,41,56], RSL change across Northwest Scotland can be described in terms of a local GIA signal caused by the growth and decay of the BIIS and a non-local GIA signal associated with the growth and decay of other ice sheets around the world. During MWP-1A, if the local GIA signal can be estimated and removed from the SLIPs, the remaining signal will be the non-local GIA signal associated with changes to the NAIS, AIS, and SIS.

Ice history and Earth rheology are not perfectly known for the British Isles. Therefore, we test 360 parameter sets when computing the local GIA signal. Specifically, we use three different BIIS models: BRITICE-CHRONO[52], PATTON2017[50,51] and ANU[2,49,79], and combine each with 120 Earth models. Because these BIIS models were constructed based on different principles (geomorphological reconstruction guided by GIA modelling for BRITICE-CHRONO, thermomechanical ice modelling for PATTON2017 and GIA modelling for ANU) they provide conservative estimates on ice history uncertainties. These ice models were combined with Earth parameters that reflect the rheological properties beneath the British Isles (denoted as near-field rheology). Specifically, we use elastic lithospheric thicknesses of 65, 72 and 80 km, upper mantle viscosities of 4, 4.5, 5, 5.5 and $6 \times 10^{20}$ Pa s and lower mantle viscosities of 1, 1.5, 2, 3, 4, 5, 7 and $10 \times 10^{22}$ Pa s. These ranges are constrained by previous GPS analysis and are largely independent of the assumed ice history[80]. We calculate the age-specific local GIA signal for each SLIP and subtract this from the original RSL reconstruction to give the non-BIIS GIA signal. The uncertainty for this procedure is considered within the inversion strategy by sampling the 360 local GIA correction values (each applied to 20,000 Monte Carlo simulations, see details below) and adding the standard deviation of the corrections to the original depth uncertainty in quadrature. After removing the local GIA signal, there is a distinct RSL jump recorded between 14.5 and 14.2 ka BP, which is consistent with the MWP-1A signal observed in far-field sea-level records (Supplementary Fig. 7).

**Spatial sea-level gradient.** Due to the large geographical spread of the SLIPs from Sunda Shelf and Northwest Scotland, there will be a non-negligible time-dependent difference in the RSL recorded at the different localities (i.e., a spatial sea-level gradient[15]). We apply a correction for this spatial gradient that enables us to determine the RSL change across MWP-1A at a single locality for each region. We quantify this gradient by testing 240 parameter combinations to incorporate uncertainties associated with ice history and Earth rheology. Specifically, we combine the two global ice models ICE6G_C[22] and ANU[2] with 120 different Earth models. The Earth models have an elastic lithospheric thickness of 60, 72 or 90 km, an upper mantle viscosity of 1, 3, 5, 6 or $7 \times 10^{20}$ Pa s, and a lower mantle viscosity of 0.1, 0.2, 0.3, 0.5, 0.6, 0.7, 0.9 or $1 \times 10^{22}$ Pa s for ICE6G_C, or 0.7, 0.9, 1, 1.5, 3, 4, 5 or $7 \times 10^{22}$ Pa s for the ANU model (we made different choices for lower mantle viscosity because the two ice models have different preference ranges). Because the local GIA signal in Northwest Scotland is removed separately we do not include the BIIS component of ICE6G_C and ANU when calculating the spatial gradient for Northwest Scotland to avoid a double correction. The ensemble mean of the 240 parameter combinations is used to determine the time-dependent data-specific spatial sea-level gradient (Supplementary Fig. 2). Site 18300 and Applecross are defined as reference sites for Sunda Shelf and Northwest Scotland respectively (red stars in Fig. 2 and Supplementary Fig. 2), to which all other sites are corrected. As for the local GIA signal correction, the uncertainty in this procedure is added to the original depth uncertainty in quadrature.

**Viscous component of sea-level change.** To correct for the viscous signal across MWP-1A, we estimate the viscous contribution of RSL change to each SLIP, accounting for their specific age and location. Given that the viscous response to ice melt prior to MWP-1A ($\Delta RSL_{PreViscous}$) will be approximately linear over MWP-1A[15,18], we quantify this linear rate by considering the viscous response to ice sheet change between the end of the last interglacial and the start of MWP-1A (14.65 ka BP). We assume no melting after 14.65 ka BP and calculate the linear rate of RSL change during the following 1,000 years (Supplementary Fig. 3). This linear rate is used to determine the $\Delta RSL_{PreViscous}(\varphi)$ signal that is specific to the age and location of each SLIP, assuming the viscous contribution is 0 at the initiation of MWP-1A (14.65 ka BP). For all sites except Northwest Scotland we use the mean

value derived from a 240-member GIA model ensemble, as described in the previous section. For Northwest Scotland, since the dominant viscous signal here relates to SIS-induced peripheral bulge subsidence (roughly 90% of the signal), which primarily depends on the local rheology of the British Isles, we use the near-field Earth parameters described in the section on Northwest Scotland. Again, to avoid a double correction, we did not include the BIIS in the global ice model when calculating $\Delta RSL_{PreViscous}(\varphi)$ for Northwest Scotland.

Because the $\Delta RSL_{DurViscous}(\varphi)$ terms depend on the ESL values in Eq. 4, which are unknown during the first iteration, we neglect this component of RSL change during the first iteration. Since these terms are relatively small compared to $\Delta RSL_{Elastic}(\varphi)$ (less than 10%), neglecting them will not significantly alter the inversion result during the first iteration. Beginning from the second iteration, we scale the ice melt geometries that are used to generate the sea-level fingerprints (main text Fig. 1) according to the bias-corrected ESL values determined in the previous iteration (details below). The pattern of $\Delta RSL_{DurViscous}(\varphi)$ is then calculated assuming a linear rate of ice melt throughout MWP-1A. A range of Earth models are used, as for $\Delta RSL_{PreViscous}(\varphi)$ above, and the ensemble mean of each set is used to determine the $\Delta RSL_{DurViscous}(\varphi)$ terms, accounting for the age and location of each SLIP (see Supplementary Fig. 4).

**Monte Carlo linear regression.** To quantify the elastic-induced local MWP-1A magnitude at each site ($\Delta RSL_{Elastic}(\varphi)$ in Eq. 4), we use a Monte Carlo (MC) linear regression technique to estimate the distribution of local sea-level rise rates recorded by selected SLIPs at that site. The MC simulation approach is used to capture the potentially asymmetric age and depth uncertainties of different types of sea-level indicators by randomly sampling each sea-level index point's depth and chronological distributions. These distributions are calculated following the methodology of Hibbert et al.[35,43].

For the coral sea-level indicators, we test two methods for representing their indicative depth distributions. First, we use an empirically-derived taxon-specific depth-habitat distribution for each coral-based sea-level indicator[43], which is obtained using the modern coral analogue from the Ocean Biogeographical Information System (www.iobis.org). Where possible, we use a spatially variable local coral depth-habitat distribution instead of a global compilation for each coral species. This method is denoted as the empirical scenario. Alternatively, we use the coral palaeo-water depth determinations (i.e., upper/lower limit of living range) from the original publications for different coral species. For this method, we assume a uniform distribution, in that the indicator may occur equally anywhere within the given range[43]. We denote this method as the uniform scenario. For non-coral SLIPs (including coral samples additionally constrained by vermetid gastropods in Tahiti), we use the indicative range or facies formation range as determined by the original authors, which is also assumed to be uniformly distributed. Furthermore, when sampling the depth distributions of all SLIPs, we considered the error distribution associated with each of the GIA corrections described above and elevation measurement uncertainties due to coring, levelling and tectonic correction if necessary. We exclude any data explicitly stated as not in situ by the original authors.

The chronological probability distributions depend on the method used to date each SLIP. For samples that are radiocarbon dated, we use OxCal version 4.4[81] to obtain the calibrated age probability distribution by recalibrating the conventional radiocarbon age and uncertainty using the latest calibration curves: IntCal20[82] for Northern Hemisphere terrestrial samples; SHCal20[83] for Southern Hemisphere terrestrial samples and Marine20[84] for all marine samples. For marine samples, we apply appropriate, updated (i.e., calculated using Marine20) local marine reservoir corrections ($\Delta R$; http://calib.org/marine). For all other samples, U-series ages have been recalculated where necessary, assuming a closed system with the latest decay constants[85]. Only U-series ages passing the following age reliability screening criteria (calcite < 2%, [$^{232}$Th] $\leq$ 2 ppb, $\delta^{234}U_{initial} = 147 \pm 5$ ‰) are considered. A normal distribution is adopted for U-series ages, whereas our radiocarbon ages use the full age probability distribution[43]. For any replicated ages, we use the inverse weighted mean value/distribution of each replicate group.

For each of our six sites, we use MC simulation to randomly sample the age and depth distributions of each selected SLIP, and for each MC sampling, we use a weighted least square method to compute an optimum straight line to fit the randomly sampled points. The slope of this line is the averaged RSL rate across MWP-1A (units m/ka), which is assumed to be temporally linear throughout MWP-1A. Note that, since the lack of temporal resolution and uneven temporal distribution of sea-level data prohibit our ability to capture the maximum rates of sea-level rise at each site, our results should be interpreted as the averaged rate of RSL change at each site across our MWP-1A time window (14.65–14.0 ka BP). Within each weighted least square calculation, the weighting factor $w$ for each SLIP is calculated by

$$w = / \sqrt{\sigma_y^2 + (dy/dx)^2 \sigma_x^2} \qquad (5)$$

where $\sigma_y$ and $\sigma_x$ are standard deviations estimated from the depth and age distributions, respectively, and $dy/dx$ is the gradient of global sea-level change at the sampled age obtained from Lambeck et al.[2]. The last term is used to convert the effective contribution of age uncertainty into depth uncertainty. Regressions that

produce a reverse slope (i.e., a sea-level fall) are excluded as implausible. The process is repeated 20,000 times (excluding the implausible iterations) to produce a distribution of local MWP-1A sea-level rise rates for each site. Lastly, the local MWP-1A magnitude is obtained by scaling this linear rate to the MWP-1A duration. We use a 500-year MWP-1A duration in this study since it leads to more comparable results with the MWP-1A magnitude from regional ice-sheet reconstructions, but our inversion results can be linearly scaled to any assumed MWP-1A duration for comparison.

**Data extrapolation**. For HYD and NOG, the SLIPs only show a rapid ~10 m sea-level rise between 14.6 and 14.4 ka BP with no clear post-MWP-1A SLIP until the initiation of new coral reefs at ~13.0 ka BP, showing another 20–25 m sea-level rise. This sequence is identified as "rapid growth then drowning and further landward migration" (Webster et al. p. 420[36]; see their Fig. 4a). In order to constrain RSL at the end of MWP–1A, we adopt a data extrapolation approach that uses the large number of SLIPs between ~13.0 and 12.0 ka BP at the two sites to extrapolate RSL backwards in time. To ensure the accuracy of the data extrapolation we only use SLIPs that pertain to a shallow, high-energy/exposed reef edge environment (the cA coral-algal assemblage[36]), i.e. SLIPs which have a relatively small depth uncertainty. The data extrapolation was implemented using the same MC linear regression method as above, in combination with the uniform scenario (Supplementary Fig. 5) as the empirical depth distribution for some SLIPs contains a bimodal habitat depth, resulting in over-large extrapolation uncertainty. The depth uncertainty of the extrapolated data point is defined by the extrapolation process, and it was assigned an age uncertainty of 0.1 ka assuming a normal distribution.

**Inversion for MWP-1A sources**. Based on the site-specific elastic-induced local MWP-1A magnitude distributions ($\Delta RSL_{Elastic}(\varphi)$) estimated above, the inversion for MWP-1A sources can be made by identifying the optimum ESL parameters in Eq. 4 for each of the 20,000 MC iterations. This is achieved using a weighted non-negative least square method using the Lawson−Hanson algorithm[53] since we assume that ice sheet growth during MWP-1A would be implausible. The weighting factor for each site is given by $w(\varphi) = 1/\sigma(\varphi)2$ where $\sigma(\varphi)$ is the standard deviation estimated from the local MWP-1A magnitude distributions.

**Jackknife resampling**. After each iteration of our method (i.e., each time we invert for the sources of MWP-1A, see Supplementary Fig. 1), we adopt a jackknife resampling technique to correct for any bias associated with data over-dependency. Specifically, we invert for the sources of MWP-1A six times, each time removing one site from the six-site database. The difference between the mean jackknife inversion result and the original inversion result is defined as the bias contained in the original inversion result. Subtracting this bias from the original result yields a bias-corrected inversion of MWP-1A sources.

**RSL prediction**. We modified the deglaciation history during MWP-1A in the ANU ice model[2] and used this revised model to predict RSL variation at the six sea-level sites used in this study. This revised model (denoted as the ANU_MWP model) was created by leaving the ice history prior to MWP-1A unchanged, but assuming that ice loss during MWP-1A followed the magnitude and spatial pattern of ice loss represented by the mean of our inversion result: 13.1 m NAIS, 3.3 m SIS, and 1.5 m AIS. The rate of ice melt during MWP-1A was assumed to be linear. In the original ANU model, there is not enough ice for the SIS to melt 3.3 m during MWP-1A. We therefore decrease the rate of ice melt between 16.5 and 14.5 ka BP (the latter is the time of MWP-1A initiation in the ANU model) to ensure there is enough ice to provide 3.3 m ESL melt during MWP-1A (Fig. 4h). The synthetic test of a larger AIS contribution to MWP-1A adopted the same pre-MWP-1A ice geometries as in the ANU_MWP model but used different MWP-1A sources: 9.6 m NAIS, 3.3 m SIS and 5 m AIS. The modified ice models were combined with the 120 ANU-specific Earth models to produce RSL curves from the last interglacial to present.

## Data availability

All sea-level data used for this study are publicly available and can be accessed from cited original publications. The datasets generated for this publication are available in the Zenodo database (https://zenodo.org/record/4459366#.YCbVyRP7TzU) with the identifier https://doi.org/10.5281/zenodo.4459366[86].

## Code availability

The code for generating the depth uncertainty distribution for each type of sea-level indicator can be accessed via https://figshare.com/articles/Matlab_Code_-_calculation_of_sea_level/5890579. The codes used to invert MWP-1A magnitude and sources are available at https://github.com/yc-lin-geo/lin_MWP1A_sources.

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

## Acknowledgements

The authors thank Chris D. Clark, Jeremy C. Ely and Henry Patton for providing their ice models, and W.R Peltier and Lev Tarasov for making their ice models publicly available. Y.L. was supported by China Scholarship Council—Durham University joint scholarship. F.D.H. received funding from the European Union's Horizon 2020 research and innovation programme under the Marie Skłodowska-Curie grant agreement (No. 838841—ExTaSea). S.L.B. was supported by the Natural Environment Research Council consortium grant BRITICE-CHRONO NE/J009768/1 and has benefited from the PalGlac team of researchers and received funding from the European Research Council (ERC) to Chris Clark under the European Union's Horizon 2020 research and innovation programme (Grant agreement No. 787263). The collection of the Scottish isolation basin data was supported by NERC Grants GST/02/0760 and GST/02/0761. The authors acknowledge PALSEA, a working group of the International Union for Quaternary Sciences (INQUA) and Past Global Changes (PAGES), which in turn received support from the Swiss Academy of Sciences and the Chinese Academy of Sciences. GIA calculations in this study were performed on the Terrawulf cluster, a computational facility supported through the AuScope Australian Geophysical Observing System (AGOS) and the Australian National Collaborative Research Infrastructure Strategy (NCRIS).

## Author contributions

Y.L. led the research; Y.L, F.D.H., P.L.W. and S.A.W conceived the scope and design of the research. Y.L., P.L.W., S.A.W and F.D.H. led the writing of the manuscript. P.L.W., A.P. and S.L.B. advised Y.L. in performing GIA modelling. F.D.H., S.A.W and I.S. advised Y.L. in sea-level and field data analysis. All authors contributed ideas and to the writing of the manuscript.

## Competing interests

The authors declare no competing interests.
