## [Peer Review File · Nature Communications]

REVIEWER COMMENTS

Reviewer #1 (Remarks to the Author):

Lin et al. present robust and compelling evidence for the ice sheet sources of the MWP-1A that will be very important also beyond the paleo sea-level and glacioisostasy research community. They provide an independent line of evidence (independent from reconstruction or modeling of individual ice sheets) of the ice sheet sources of the MWP-1a that has riddled the paleoclimate and paleo ice sheet/sea-level research communities for decades.

Specifically, they employ an improved methodology to quantify local sea-level rise from a combination of near, intermediate and far field sites during the MWP-1A and use this to invert the local sea-level fingerprint into a probabilistic estimate of the ice sheet contribution from the different ice sheets. These sites are significantly better in discriminating between meltwater sources than the far-intermediate field sites utilized in previous sea-level fingerprinting exercises. The main finding is that the MWP-1A was sourced from a dominant North American, a major Scandinavian and a minor Antarctic melt contribution.

Different from previous sea-level fingerprinting this study present evidence for that meltwater from Northern Hemisphere ice sheets by far was the dominant source for the MWP-1A and show that a massive flux of freshwater was routed to the North Atlantic, the Nordic Seas and the Arctic Ocean at the same time as the relatively warm Bølling interstadial when the AMOC was relatively strong (McManus et al., 2004), pointing to that the ocean circulation response to meltwater forcing is not as straight forward as meltwater hosing model experiments suggest.

With this new independent quantification of meltwater forcing to the North Atlantic region, the modeling and paleoclimate communities can put more confidence in using ice sheet reconstructions and modeling (e.g. Brendryen et al., 2020; Tarasov et al., 2012; Gregoire et al., 2012) as forcing for modeling experiments that will lead to improved understanding of the climate system response to melting ice sheets. Improved understanding of meltwater impact on North Atlantic oceanic circulation is of immense importance for our ability to project the future fate of the AMOC in a scenario with increased melting of the Greenland ice sheet in a warming world.

While I find the evidence and how it is presented convincing and without fatal flaws, there are some things I would like the authors to clarify or further address:

1. Reported meltwater contribution. It is not clear to me where the numbers for each meltwater source reported in the abstract come from. For the Scandinavian ice sheet, the max and min range seems to come from the 95% interquantile range of the filtered probability distributions in Fig. 5, while the central estimate of 3.3 m (seemingly coming from the unfiltered results in Fig. 3b) is much lower than the p-max/median of the finger-print inversion in Fig. 5 (about 4.5 m). For consistency and clarity, I suggest that the authors report the median and the 95% range of the probability distribution of the inverted and filtered sea-level fingerprints as well as (from Fig. 5).

2. Filtering of the sea-level fingerprint inversion. While there are good arguments for filtering out solutions that will result in a sea-level oscillation in Scotland (i.e. an Antarctic contribution of more than ca 5 m) which will lead to sea-level histories in Scotland that are inconsistent with the data, I find it less clear why solutions where the SIS contribute more than 6 m of sea-level rise during MWP-1A have been filtered out. The reasoning for the authors to do so was that it "...reduced over-dependence on the SIS contribution when seeking to fit the Northwest Scotland data." (line 329-330) What does this mean? Would the inversion results where all fingerprint sites are included be much different from those presented in Fig. 5 if this filter was not used (but with the "Antarctic > 5 m" filter included)? Also, a justification for the use of the "SIS > 6 m" filter is that a 6 m meter contribution is a conservative estimate as the current SIS reconstructions give in the range of 1.5-

5.5 m SLE (I suppose these numbers refer to ice above flotation). By imposing this filter, the result from the fingerprinting exercise will, however, not be fully independent from individual ice sheet reconstructions anymore.

3. Brendryen et al. (2020) SIS meltwater contribution. In line 351-354 it is stated that Brendryen et al. (2020) suggest that "...the southern Barents Sea sector may collapse during MWP-1A, contributing 2.1-5.5 m to GMSL rise (only considering the eustatic contribution from grounded ice above flotation)". It is unclear to me how the authors came to the amount of ice above flotation as Brendryen et al. (2020) explicitly state that the reported SLE is from ice both above and below flotation. It is also unclear to me whether this number only regard the Barents Sea or if it includes both the Scandinavian and the Barents-Svalbard ice sheets. Please clarify this.

4. Freshwater forcing from the SIS. In line no. 359 it is stated that a freshwater flux of 0.08 Sv was routed to the Nordic Sea from the SIS during MWP-1A. Is this number derived from lost ice above flotation? Would it not be more relevant as ocean buoyancy forcing if the stated number refers to the meltwater flux from all ice lost from the SIS (both below and above flotation) during MWP-1A?

Reviewer #2 (Remarks to the Author):

This is a **really** interesting and useful study that I strongly support for publication in Nature Communications (albeit with the caveat that I am not an expert in GIA or sea level [fingerprinting]). Essentially, the presented results suggest that published transient simulations of the last deglaciation and, specifically of the Bolling Warming/MWP1a period are 'wrong' since a large Northern Hemisphere Source of MWP1a fairly consistently induces cooling in climate models, whereas studies by Liu et al. (2009), Menviel et al. (2010) require Northern Hemisphere melting to reduce/stop, or others require it to be very slow (Yeung et al., 2019; Obase and Abe-Ouchi, 2019) around the time of MWP1a. Not just an interesting conundrum, but invaluable evidence for model experiment protocols (e.g. Ivanovic et al., 2016) – I think these authors and those implementing the protocol would hugely benefit from the results presented in this manuscript. As the authors of this reviewed manuscript point out, hitherto published evidence has several non-unique possible solutions for the origin of MWP1a. Thus, in providing a more concrete scenario, the presented work could be a big step forward in directing conceptual and numerical model studies, and therefore for finding more robust mechanisms to explain the last deglaciation chain of events and evaluate model performance/sensitivity to freshwater forcing. I am very excited about and highly supportive of this work.

The manuscript is already very well polished, easy to read/understand and I recommend it for publication either in its present form, or after incorporating the following minor comments. These comments are mainly subjective, but would improve the framing of the results in the context of ice sheet/climate modelling studies, facilitating the uptake of the proposed scenario in new model experiments. The framing of the sea-level fingerprinting (challenges to date, and presented solutions to overcome) is excellent – very clear, on topic and highly educational [at least to me]. All figures are superb and highly informative, Figures 3 and 5 is particularly useful for the type of application of these results suggested in the first paragraph of this review (above)..

As mentioned, I am not sufficiently expert in the new records, chronologies, or presented modelling techniques to comment on these aspects, although I found the relevant text and associated figures very well presented and easy to follow.

MINOR/DETAILED COMMENTS

Line 4-6: 'It was driven by the collapse of vulnerable ice sheet sectors, forced by rapid Northern

Hemispheric warming and disruptions in oceanic and atmospheric circulation^{3,4}. Ice sheet collapse may well have been driven by rapid warming, but this is not necessarily so, e.g. Gregoire et al. 2012, 2016.

Line 6-8 (and general framing): 'The ice-climate feed backs [should be 'feedbacks?'] operating during this period provide a vital analogue for future sea-level rise and climate change,' I agree it is interesting, but the statement that it is a 'vital analogue' needs justifying when so much is different between then and now. Fundamentally, how is it an 'analogue' and why is it 'vital'? E.g. how is continental North American ice sheet collapse a vital analogue for [near?] future ice sheet change (or do the authors include future millennia in this timeframe of 'future' – perhaps clarify here?)? I find this hard to envisage. It would require there to be mechanistic similarities and/or similarities in the geometries of the ice sheets and climate/ice sheet forcings - please elaborate on what, where and when these are, or be more precise in the framing (e.g. focus on the fact that it is useful/interesting to understand the mechanisms of change/interaction). Perhaps also more to the point, it is a really interesting question to understand the provenance of MWP1a because the climate/ocean impacts *may* be opposing, depending at least in part on the size of the Antarctic (and Northern Hemisphere) contribution(s) (e.g. Ivanovic et al., 2017, 2018; Menviel et al., 2010; Stouffer et al., 2007; Swingedouw et al., 2009; Weaver et al., 2003; Yeung et al. 2019), which has significant implications for understanding and simulating the chain of events observed in the last deglaciation (e.g. Liu et al., 2009; Menviel et al., 2010; Obase and Abe-Ouchi et al., 2020) – see comment above. I would therefore suggest expanding the opening few paragraphs (and relevant abstract, results/discussion/conclusions sections, as the authors see fit) to justify (or otherwise remove) the 'vital analogue' argument and instead refocus the manuscript to really drill home the more robust point that the presented work indicates we very much do not understand the climate evolution of the mid-deglaciation and therefore by implication we have incomplete knowledge of (and may not be able to simulate) the fundamental climate-ice-ocean interactions involved. Fine to then add the relevance to future climate/ice/sea level *if* properly explained.

Line 399-402: 'Use of our results to prescribe the global pattern of meltwater discharge during MWP-1A may yield novel insights into the sequencing of ice-ocean-climate interactions during this recent abrupt climate change event.' I'm sure it is already evident in my review that I strongly agree with this statement – the presented results provide firm evidence to hugely narrow the scope of uncertainty for what glaciologists/climatologists should be testing/prescribing/seeking to explain. Not only that, but it casts existing results (as commented above) in an interesting and informative light. This is exactly the sort of work that we desperately need. Thank you.

REFERENCES CITED IN REVIEW (noting that most of these are already included in the manuscript)

Gregoire, L. J., Payne, A. J. and Valdes, P. J.: Deglacial rapid sea level rises caused by ice-sheet saddle collapses, *Nature*, 487(7406), 219–222, doi:10.1038/nature11257, 2012.

Gregoire, L. J., Otto-Bliesner, B., Valdes, P. J. and Ivanovic, R. F.: Abrupt Bølling warming and ice saddle collapse contributions to the Meltwater Pulse 1a rapid sea level rise, *Geophys. Res. Lett.*, 43(17), 2016GL070356, doi:10.1002/2016GL070356, 2016.

Ivanovic, R. F., Gregoire, L. J., Kageyama, M., Roche, D. M., Valdes, P. J., Burke, A., Drummond, R., Peltier, W. R. and Tarasov, L.: Transient climate simulations of the deglaciation 21–9 thousand years before present (version 1) – PMIP4 Core experiment design and boundary conditions, *Geosci. Model Dev.*, 9(7), 2563–2587, doi:10.5194/gmd-9-2563-2016, 2016.

Ivanovic, R. F., Gregoire, L. J., Wickert, A. D., Valdes, P. J. and Burke, A.: Collapse of the North American ice saddle 14,500 years ago caused widespread cooling and reduced ocean overturning circulation, *Geophys. Res. Lett.*, 44(1), 383–392, doi:10.1002/2016GL071849, 2017.

Ivanovic, R. F., Gregoire, L. J., Wickert, A. D. and Burke, A.: Climatic Effect of Antarctic Meltwater Overwhelmed by Concurrent Northern Hemispheric Melt, *Geophysical Research Letters*, 45(11), 5681–5689, doi:10.1029/2018GL077623, 2018.

Liu, Z., Otto-Bliesner, B. L., He, F., Brady, E. C., Tomas, R., Clark, P. U., Carlson, A. E., Lynch-Stieglitz, J., Curry, W., Brook, E., Erickson, D., Jacob, R., Kutzbach, J. and Cheng, J.: Transient Simulation of Last Deglaciation with a New Mechanism for Bølling-Allerød Warming, *Science*, 325(5938), 310–314, doi:10.1126/science.1171041, 2009.

Menviel, L., Timmermann, A., Timm, O. E. and Mouchet, A.: Climate and biogeochemical response to a rapid melting of the West Antarctic Ice Sheet during interglacials and implications for future climate, *Paleoceanography*, 25(4), PA4231, doi:10.1029/2009PA001892, 2010.

Menviel, L., Timmermann, A., Timm, O. E. and Mouchet, A.: Deconstructing the Last Glacial termination: the role of millennial and orbital-scale forcings, *Quaternary Science Reviews*, 30(9–10), 1155–1172, doi:10.1016/j.quascirev.2011.02.005, 2011.

Obase, T. and Abe-Ouchi, A.: Abrupt Bølling-Allerød Warming Simulated under Gradual Forcing of the Last Deglaciation, *Geophysical Research Letters*, 46(20), 11397–11405, doi:10.1029/2019GL084675, 2019.

Stouffer, R. J., Seidov, D. and Haupt, B. J.: Climate Response to External Sources of Freshwater: North Atlantic versus the Southern Ocean, *J. Climate*, 20(3), 436–448, doi:10.1175/JCLI4015.1, 2007.

Swingedouw, D., Fichefet, T., Goosse, H. and Loutre, M. F.: Impact of transient freshwater releases in the Southern Ocean on the AMOC and climate, *Clim Dyn*, 33(2–3), 365–381, doi:10.1007/s00382-008-0496-1, 2009.

Weaver, A. J., Saenko, O. A., Clark, P. U. and Mitrovica, J. X.: Meltwater Pulse 1A from Antarctica as a Trigger of the Bølling-Allerød Warm Interval, *Science*, 299(5613), 1709–1713, doi:10.1126/science.1081002, 2003.

Yeung, N. K. H., Menviel, L., Meissner, K. J. and Sikes, E.: Assessing the Spatial Origin of Meltwater Pulse 1A Using Oxygen-Isotope Fingerprinting, *Paleoceanography and Paleoclimatology*, 34(12), 2031–2046, doi:10.1029/2019PA003599, 2019.

Reviewer #3 (Remarks to the Author):

Review of Lin et al.

The study by Lin et al. uses multiple sites with sea-level records spanning Meltwater Pulse 1a (MWP-1a) to fingerprint the source of ice sheet melting using glacial isostatic adjustment modeling. Through a statistically-sophisticated treatment of sea-level index points and their associated uncertainties in locations across the globe, the authors invert for MWP-1a ice sheet source, by

optimizing contributions from continental ice sheets to fit the relative sea level data at these sites. The authors conclude that only a North American dominant scenario is permitted by their observational constraints, and an Antarctic scenario is precluded. This is an excellent and rigorous study that is also well written and interesting. The results are novel, and weigh in on a topic of considerable interest to the sea-level community and Earth scientists more broadly.

I have a one main suggestion and other minor comments. The authors emphasize the importance of including near field sites to have more differentiating power between different ice melting scenarios. Since the Scotland and Barbados sites are sensitive to their respective nearby ice sheet, I am curious about whether the authors would be able to test different geometries of melt in these ice sheets. My understanding is that the authors used the same ice melt geometry for each ice sheet and scaled these up or down to fit the relative sea level observations. However, there are obviously large uncertainties in the geographic pattern of ice melting over this time. I think it would be fascinating to explore this uncertainty, and change the location or spatial pattern of ice melt, especially in the North American and Scandinavian ice sheets, which the two near field sites are most sensitive to.

In particular, the North American ice sheets should be considered separately. A recent study suggested that the ice sheet saddle did not contribute to MWP-1a, and rather melted primarily during the Younger Dryas, based on sea level records at the Bering Strait. In this study, as in many prior ice histories, the saddle region melts considerably during MWP-1a. However, it is not clear as the study has been done, whether the Barbados record requires this geometry or if a substantial melt in the eastern Laurentide would similarly fit observations. I think performing tests that separate the eastern and western regions of the North American Ice Sheets would allow this study greater influence in weighing in on this debate about ice sheet sources at MWP-1a. At the very least, the idea of different ice melt geometries (and the limitations of using a single ice melt geometry) should be brought in during the discussion.

Line by line comments

Line 4: I believe it is not clear that ice sheet melt is driven by Northern Hemisphere warming given that Bolling Allerod is 500 yrs before MWP1a, which seems an ample lag in forcing and response. I think this is a subject of much debate, and should probably be written here as a suggestion rather than fact.

Line 26: Should cite Clark, P. U. et al. (2002) 'Sea-level fingerprinting as a direct test for the source of global meltwater pulse IA.', *Science* (New York, N.Y.), 295(5564), pp. 2438–2441. doi: 10.1126/science.1068797. here, since this is the first study to fingerprint MWP-1a. Not sure why this is only cited in the previous sentence.

Figure 1: The only near field site is Scotland. How might these results look different with other near field sites? As in, what is the possibility that there is a range of options that would fit Scotland relatively well, if other sites could add differentiating power?

Although not at MWP1a, the recent Pico et al., *Sci Adv*, 2020 study used sea level at the Bering Strait to fingerprint melt sources during the Younger Dryas, and concluding that a large portion of the western North American ice sheets melted during this time. They argue that this result precludes the western North American ice sheets from contributing substantially to MWP-1a (which possibly contradicts the results in this study, depending on the geometry of ice melt needed to fit records in Scotland and Barbados).

Line 93: Wording is a bit confusing, I think it would make sense to start with something like the sentence on line 100 "We determine all three corrections...", because I felt confused on how the corrections were done, but then this was explained after.

Line 109: What was the requirement for sea level records used in this study? Is there a certain robustness for chronology across MWP-1a that each site must have?

I realize this is a fingerprinting study, but I wonder if the authors think it would be valuable to look at a longer timescale for RSL records (perhaps this would expand the number of records, which could include more near field sites) and consider the longer viscoelastic sea level response to the inferred ice melting scenario? I realize this is outside the scope of the present study, however I think this could potentially be highlighted as a future direction or it could be commented on why this elastic fingerprinting method is more appropriate.

Figure 2: It would be helpful to label HYD and NOG by the country they are in (or Great Barrier Reef) for easier reference. The other geographic locations are more intuitive.

Figure 2: Why are there sometimes clouds of probability above the data points?

Figure 3: Is the right side of the y axis meters of GMSL? If so this should be labeled. If not, I am not sure what this represents. I also think the "local MWP-1a magnitude" should perhaps be "local MWP-1a relative sea-level change magnitude", if I am correct in understanding that the number represents the total change in RSL over MWP-1a at each site.

Figure 4: Since this figure does not compare RSL predictions using the different ice model modifications (ANU vs ANU-MWP), I'm not sure as is that it helps the reader understand why the best fit model is actually an improved fit. Perhaps the RSL predictions with the standard ANU model can be shown as a comparison? Or a misfit using the two RSL predictions?

For the Scottish site, it might be helpful to label on the figure "stratigraphy shows no oscillation" since this site requires a different interpretation from the others.

Line 254: What about considering different parts of ice sheets? For example eastern vs western North American ice sheets? Geometry of ice melt is important for the elastic fingerprint, although these sites might be far away enough from ice sheets that it doesn't matter. Would be interesting to see sensitivity to ice sheet geometry.

Line 372: I would disagree with this sentence. The saddle-collapse mechanism has been modeled with dynamic ice sheet simulations forced with a deglacial climate, however the timing of saddle collapse in this modeling is not MWP-1a. Rather Gregoire et al. attribute a mechanism for a rapid melting of the North American ice sheets with MWP-1a. In model time, the timing is closer to Younger Dryas, which suggests that the relationship of an ice saddle collapse with Bolling Allerod warming is not so clear.

Indeed Pico et al. 2020 suggest that the North American saddle melted primarily during the Younger Dryas from 13-11.5 ka, rather than during MWP-1a, based on fingerprinting the flooding history of the Bering Strait. The study explores different MWP-1a melting scenarios in the supplement, and finds that most other combinations of ice sheet melt would fit the Bering Strait record. However, this result requires that ice melt during MWP-1a in North America is sourced from the eastern Laurentide, and not the ice sheet saddle, as suggested by Gregoire et al. This Pico et al. study should perhaps be brought into the discussion to frame the debate around potential meltwater sources for MWP-1a.

Pico, T., Mitrovica, J. X. and Mix, A. C. (2020) 'Sea level fingerprinting of the Bering Strait flooding history detects the source of the Younger Dryas climate event', *Science Advances*.

Consequently, I am curious whether the sensitivity of Barbados to North American ice sheets could differentiate between eastern and western sources. Would it be possible to test this by treating these two areas separately? It would be interesting to learn whether the data at Barbados requires

ice melt from a certain region of the North American ice sheets, and whether the results in this paper are consistent or not with the conclusions drawn in Pico et al.

Figure S5: I think that the ice melt geometries are very important, and should at least be described in the main text. After reading the main text, I was unsure if the ice melting scenarios were uniform ice thicknesses or not. These figures are helpful, but if it is not possible to include them, it would be helpful to describe in some way that the ice histories used are characterized by spatially heterogeneous melting, and perhaps even explaining what regions contain the most ice melt within these different ice sheets.

REVIEWER COMMENTS

Reviewer #1 (Remarks to the Author):

Lin et al. present robust and compelling evidence for the ice sheet sources of the MWP-1A that will be very important also beyond the paleo sea-level and glacioisostasy research community. They provide an independent line of evidence (independent from reconstruction or modeling of individual ice sheets) of the ice sheet sources of the MWP-1a that has riddled the paleoclimate and paleo ice sheet/sea-level research communities for decades.

Specifically, they employ an improved methodology to quantify local sea-level rise from a combination of near, intermediate and far field sites during the MWP-1A and use this to invert the local sea-level fingerprint into a probabilistic estimate of the ice sheet contribution from the different ice sheets. These sites are significantly better in discriminating between meltwater sources than the far-intermediate field sites utilized in previous sea-level fingerprinting exercises. The main finding is that the MWP-1A was sourced from a dominant North American, a major Scandinavian and a minor Antarctic melt contribution.

Different from previous sea-level fingerprinting this study present evidence for that meltwater from Northern Hemisphere ice sheets by far was the dominant source for the MWP-1A and show that a massive flux of freshwater was routed to the North Atlantic, the Nordic Seas and the Arctic Ocean at the same time as the relatively warm Bølling interstadial when the AMOC was relatively strong (McManus et al., 2004), pointing to that the ocean circulation response to meltwater forcing is not as straight forward as meltwater hosing model experiments suggest.

With this new independent quantification of meltwater forcing to the North Atlantic region, the modeling and paleoclimate communities can put more confidence in using ice sheet reconstructions and modeling (e.g. Brendryen et al., 2020; Tarasov et al., 2012; Gregoire et al., 2012) as forcing for modeling experiments that will lead to improved understanding of the climate system response to melting ice sheets. Improved understanding of meltwater impact on North Atlantic oceanic circulation is of immense importance for our ability to project the future fate of the AMOC in a scenario with increased melting of the Greenland ice sheet in a warming world.

We thank the reviewer for recognizing the importance of this study for better understanding the climate system. We address this reviewer's specific comments, below. Our responses are in blue.

While I find the evidence and how it is presented convincing and without fatal flaws, there are some things I would like the authors to clarify or further address:

1. Reported meltwater contribution. It is not clear to me where the numbers for each meltwater source reported in the abstract come from. For the Scandinavian ice sheet, the max and min range seems to come from the 95% interquantile range of the filtered probability distributions in Fig. 5, while the central estimate of 3.3 m (seemingly coming from the unfiltered results in Fig. 3b) is much lower than the p-max/median of the finger-print inversion in Fig. 5 (about 4.5 m). For consistency and clarity, I suggest that the authors report the median and the 95% range of the probability distribution of the inverted and filtered sea-

level fingerprints as well as (from Fig. 5).

We apologize for this confusion. We originally reported mean results without the constraint imposed by assuming no sea-level oscillation in Scotland. We now report median values from the inverted and filtered results, as suggested.

2. Filtering of the sea-level fingerprint inversion. While there are good arguments for filtering out solutions that will result in a sea-level oscillation in Scotland (i.e. an Antarctic contribution of more than ca 5 m) which will lead to sea-level histories in Scotland that are inconsistent with the data, I find it less clear why solutions where the SIS contribute more than 6 m of sea-level rise during MWP-1A have been filtered out. The reasoning for the authors to do so was that it "...reduced over-dependence on the SIS contribution when seeking to fit the Northwest Scotland data." (line 329-330) What does this mean? Would the inversion results where all fingerprint sites are included be much different from those presented in Fig. 5 if this filter was not used (but with the "Antarctic > 5 m" filter included)? Also, a justification for the use of the "SIS > 6 m" filter is that a 6 m meter contribution is a conservative estimate as the current SIS reconstructions give in the range of 1.5-5.5 m SLE (I suppose these numbers refer to ice above flotation). By imposing this filter, the result from the fingerprinting exercise will, however, not be fully independent from individual ice sheet reconstructions anymore.

The statement about 'over-dependence on the SIS contribution' related to the strong sensitivity of our inversion results on the Northwest Scotland data. This text has now been removed, and following the reviewer's suggestion, we have recalculated the inversion uncertainty ranges without applying a limit on the SIS contribution. A comparison of inversion results with and without this SIS constraint is shown below:

	With 6m SIS constraint	Without 6m SIS constraint
	95% CI (mean)	95% CI (mean)
Total	15.6-19.9 (17.9)	15.7 – 20.2 (17.9)
NAIS	6.3-15.4 (12.4)	5.6-15.4 (12.0)
AIS	0-5.4 (1.1)	0-5.9 (1.3)
SIS	3.2-5.6(4.5)	3.2-6.4 (4.6)

Without applying the SIS constraint, the overall results do show a larger uncertainty range as expected but the uncertainty ranges, as well as the median value, are similar to the results with the constraint. This is largely due to the inclusion of the Barbados site, which helps to prevent the over-dependence problem. We have revised the inversion results reported in the manuscript and no longer apply the 6m SIS constraint, this maintains the independence of our results from previously published ice sheet reconstructions. We thank the reviewer for inspiring this improvement.

3. Brendryen et al. (2020) SIS meltwater contribution. In line 351-354 it is stated that Brendryen et al. (2020) suggest that "...the southern Barents Sea sector may collapse during MWP-1A, contributing 2.1-5.5 m to GMSL rise (only considering the eustatic contribution from grounded ice above flotation)". It is unclear to me how the authors came to the amount of ice above flotation as Brendryen et al. (2020) explicitly state that the reported SLE is from ice both above and below flotation. It is also unclear to me whether this number only regard the Barents Sea or if it includes both the Scandinavian and the Barents-Svalbard ice sheets.

Please clarify this.

We thank the reviewer for querying this point. The values reported by Brendryen et al. (2020) are sea-level equivalent (SLE) values. They are derived by calculating the ice-volume change across MWP-1A and then dividing by the area of the ocean. The ice-volume change calculation includes ice that lies below hydrostatic equilibrium, which will not contribute to global mean sea-level rise. We wish to report the eustatic contribution to sea level, i.e. the contribution from ice above hydrostatic equilibrium, and so we seek to estimate and subtract the volume of ice below flotation from the Brendryen et al. (2020) values.

The Brendryen et al. (2020) calculations are partially based on the ice sheet area-volume relationship of Patton et al. (2016; 2017 – referred to here as PATTON2017). Using the PATTON2017 model as the ice sheet input to a GIA model we are able to quantify the Barents Sea sector collapse induced eustatic sea level rise as well as the total change in ice volume. The difference between these quantities – equivalent to ~0.5 m sea-level change – is the magnitude of the correction that must be applied to the Brendryen et al. (2020) values. Subtracting 0.5 m from the Brendryen et al. (2020) values (4.5-7.9 m SLE) yields a eustatic contribution of 4.0-7.4 m. Note that this range is slightly different to that reported in the original version of our manuscript because we previously used the incorrect value for the change in SIS ice volume. The updated range and a brief summary of our method are reported in the revised version of our manuscript (lines 229-230 in track change word file).

We refer to the combined Barents-Svalbard and Scandinavian Ice Sheets together, a clarification “accompanied by marginal retreat of the Scandinavian Ice Sheet,” has been added to line 228 (in track change word file).

4. Freshwater forcing from the SIS. In line no. 359 it is stated that a freshwater flux of 0.08 Sv was routed to the Nordic Sea from the SIS during MWP-1A. Is this number derived from lost ice above flotation? Would it not be more relevant as ocean buoyancy forcing if the stated number refers to the meltwater flux from all ice lost from the SIS (both below and above flotation) during MWP-1A?

We originally only reported the contribution from ice above flotation. We agree that a freshwater flux value should also include the contribution from ice below flotation. We have updated the freshwater flux value to include ice below flotation.

Reviewer #2 (Remarks to the Author):

This is a *really* interesting and useful study that I strongly support for publication in Nature Communications (albeit with the caveat that I am not an expert in GIA or sea level [fingerprinting]). Essentially, the presented results suggest that published transient simulations of the last deglaciation and, specifically of the Bolling Warming/MWP1a period are ‘wrong’ since a large Northern Hemisphere Source of MWP1a fairly consistently induces cooling in climate models, whereas studies by Liu et al. (2009), Menviel et al. (2010) require Northern Hemisphere melting to reduce/stop, or others require it to be very slow (Yeung et al., 2019; Obase and Abe-Ouchi, 2019) around the time of MWP1a. Not just an interesting conundrum, but invaluable evidence for model experiment protocols (e.g. Ivanovic et al., 2016) – I think these authors and those implementing the protocol would hugely benefit from the results presented in this manuscript. As the authors of this reviewed manuscript point out, hitherto published evidence has several non-unique possible solutions for the origin of MWP1a. Thus, in providing a more concrete scenario, the presented work could be a big step forward in directing conceptual and numerical model studies, and therefore for finding more robust mechanisms to explain the last deglaciation chain of events and evaluate model performance/sensitivity to freshwater forcing. I am very excited about and highly supportive of this work.

The manuscript is already very well polished, easy to read/understand and I recommend it for publication either in its present form, or after incorporating the following minor comments. These comments are mainly subjective, but would improve the framing of the results in the context of ice sheet/climate modelling studies, facilitating the uptake of the proposed scenario in new model experiments. The framing of the sea-level fingerprinting (challenges to date, and presented solutions to overcome) is excellent – very clear, on topic and highly educational [at least to me]. All figures are superb and highly informative, Figures 3 and 5 is particularly useful for the type of application of these results suggested in the first paragraph of this review (above)..

As mentioned, I am not sufficiently expert in the new records, chronologies, or presented modelling techniques to comment on these aspects, although I found the relevant text and associated figures very well presented and easy to follow.

We thank the reviewer for the positive and constructive review. We address this reviewer’s specific comments below.

MINOR/DETAILED COMMENTS

Line 4-6: ‘It was driven by the collapse of vulnerable ice sheet sectors, forced by rapid Northern Hemispheric warming and disruptions in oceanic and atmospheric circulation^{3,4}’. Ice sheet collapse may well have been driven by rapid warming, but this is not necessarily so, e.g. Gregoire et al. 2012, 2016.

We thank the reviewer for pointing this out. We have changed our introductory paragraph accordingly. We have modified this sentence to “It was driven by the collapse of vulnerable ice sheet sectors, and was concurrent with rapid Northern Hemispheric warming and disruptions in oceanic and atmospheric circulation.” (line 24 in track change word file). So it is a statement of palaeoclimate condition instead of linking MWP-1A to these palaeoclimate events.

Line 6-8 (and general framing): ‘The ice-climate feed backs [should be ‘feedbacks’?] operating during this period provide a vital analogue for future sea-level rise and climate change,’ I agree it is interesting, but the statement that it is a ‘vital analogue’ needs justifying when so much is different between then and now. Fundamentally, how is it an ‘analogue’ and why is it ‘vital’? E.g. how is continental North American ice sheet collapse a vital analogue for [near?] future ice sheet change (or do the authors include future millennia in this timeframe of ‘future’ – perhaps clarify here?)? I find this hard to envisage. It would require there to be mechanistic similarities and/or similarities in the geometries of the ice sheets and climate/ice sheet forcings - please elaborate on what, where and when these are, or be more precise in the framing (e.g. focus on the fact that it is useful/interesting to understand the mechanisms of change/interaction). Perhaps also more to the point, it is a really interesting question to understand the provenance of MWP1a because the climate/ocean impacts may be opposing, depending at least in part on the size of the Antarctic (and Northern Hemisphere) contribution(s) (e.g. Ivanovic et al., 2017, 2018; Menviel et al., 2010; Stouffer et al., 2007; Swingedouw et al., 2009; Weaver et al., 2003; Yeung et al. 2019), which has significant implications for understanding and simulating the chain of events observed in the last deglaciation (e.g. Liu et al., 2009; Menviel et al., 2010; Obase and Abe-Ouchi et al., 2020) – see comment above. I would therefore suggest expanding the opening few paragraphs (and relevant abstract, results/discussion/conclusions sections, as the authors see fit) to justify (or otherwise remove) the ‘vital analogue’ argument and instead refocus the manuscript to really drill home the more robust point that the presented work indicates we very much do not understand the climate evolution of the mid-deglaciation and therefore by implication we have incomplete knowledge of (and may not be able to simulate) the fundamental climate-ice-ocean interactions involved. Fine to then add the relevance to future climate/ice/sea level if properly explained.

We agree with the reviewer that MWP-1A may not be an analogue for modern global warming due to distinct differences in ice sheet volume and distribution. We have modified our introduction accordingly to “The ice-ocean-climate feedbacks operating during this period are not well understood largely due to a lack of consensus on the sources of MWP-1A, which, in turn, were likely a key driver in stimulating rapid deglacial climate change.”, it now focuses more on understanding the mechanisms of climate change and climate-ocean interaction. Following this modification, we have also referenced more climate modelling studies to support the statement above.

Line 399-402: ‘Use of our results to prescribe the global pattern of meltwater discharge during MWP-1A may yield novel insights into the sequencing of ice-ocean-climate interactions during this recent abrupt climate change event.’ I’m sure it is already evident in my review that I strongly agree with this statement – the presented results provide firm evidence to hugely narrow the scope of uncertainty for what glaciologists/climatologists should be testing/prescribing/seeking to explain. Not only that, but it casts existing results (as commented above) in an interesting and informative light. This is exactly the sort of work that we desperately need. Thank you.

We thank the reviewer again for this encouraging comment. Since this study was also designed with the aim of helping to understand climate mechanisms during this period, we are very pleased to receive this feedback.

Reviewer #3 (Remarks to the Author):

The study by Lin et al. uses multiple sites with sea-level records spanning Meltwater Pulse 1a (MWP-1a) to fingerprint the source of ice sheet melting using glacial isostatic adjustment modeling. Through a statistically-sophisticated treatment of sea-level index points and their associated uncertainties in locations across the globe, the authors invert for MWP-1a ice sheet source, by optimizing contributions from continental ice sheets to fit the relative sea level data at these sites. The authors conclude that only a North American dominant scenario is permitted by their observational constraints, and an Antarctic scenario is precluded. This is an excellent and rigorous study that is also well written and interesting. The results are novel, and weigh in on a topic of considerable interest to the sea-level community and Earth scientists more broadly.

We thank the reviewer for this positive comment. We address this reviewer's specific comments, below.

I have a one main suggestion and other minor comments. The authors emphasize the importance of including near field sites to have more differentiating power between different ice melting scenarios. Since the Scotland and Barbados sites are sensitive to their respective nearby ice sheet, I am curious about whether the authors would be able to test different geometries of melt in these ice sheets. My understanding is that the authors used the same ice melt geometry for each ice sheet and scaled these up or down to fit the relative sea level observations. However, there are obviously large uncertainties in the geographic pattern of ice melting over this time. I think it would be fascinating to explore this uncertainty, and change the location or spatial pattern of ice melt, especially in the North American and Scandinavian ice sheets, which the two near field sites are most sensitive to.

We thank the review for pointing out this uncertainty, which is really important. In our original manuscript, we only briefly mentioned the geometry uncertainty of the Scandinavian Ice Sheet (SIS) in line 383 (original version) and did not discuss the uncertainty associated with the North American Ice Sheet (NAIS). We have now carried out some additional experiments to incorporate these uncertainties into our discussion. Please see the details of these experiments in our response to specific comments below.

In particular, the North American ice sheets should be considered separately. A recent study suggested that the ice sheet saddle did not contribute to MWP-1a, and rather melted primarily during the Younger Dryas, based on sea level records at the Bering Strait. In this study, as in many prior ice histories, the saddle region melts considerably during MWP-1a. However, it is not clear as the study has been done, whether the Barbados record requires this geometry or if a substantial melt in the eastern Laurentide would similarly fit observations. I think performing tests that separate the eastern and western regions of the North American Ice Sheets would allow this study greater influence in weighing in on this debate about ice sheet sources at MWP-1a. At the very least, the idea of different ice melt geometries (and the limitations of using a single ice melt geometry) should be brought in during the discussion.

We have carried out some additional experiments to try to separate the contribution of the Eastern and Western Laurentide Ice Sheets (ELIS and WLIS) to MWP-1A. Please see the details of these experiments in our response to specific comments below.

Line by line comments

Line 4: I believe it is not clear that ice sheet melt is driven by Northern Hemisphere warming given that Bolling Allerod is 500 yrs before MWP1a, which seems an ample lag in forcing and response. I think this is a subject of much debate, and should probably be written here as a suggestion rather than fact.

We have modified this sentence to “It was driven by the collapse of vulnerable ice sheet sectors, and was concurrent with rapid Northern Hemispheric warming and disruptions in oceanic and atmospheric circulation.” (line 23 in track change word file). So it is a statement of palaeoclimate condition instead of linking MWP-1A to these palaeoclimate events.

Line 26: Should cite Clark, P. U. et al. (2002) ‘Sea-level fingerprinting as a direct test for the source of global meltwater pulse IA.’, *Science* (New York, N.Y.), 295(5564), pp. 2438–2441. doi: 10.1126/science.1068797. here, since this is the first study to fingerprint MWP-1a. Not sure why this is only cited in the previous sentence.

We have amended this as suggested.

Figure 1: The only near field site is Scotland. How might these results look different with other near field sites? As in, what is the possibility that there is a range of options that would fit Scotland relatively well, if other sites could add differentiating power?

There are clearly a range of scenarios that will fit the Scotland RSL data. If more near-field sites were available and could be used in our analysis, the results would be different depending on two factors:

- 1) The sensitivity of the field site locations to the meltwater sources. Any additional near-field sites located near to the NAIS, SIS or AIS would be sensitive to both near-field and far-field meltwater sources and would therefore should add additional differentiating power.
- 2) The uncertainty within the near-field site RSL data. The Scotland data are really special because the local British-Irish Ice Sheet (BIIS) has largely disappeared prior to MWP-1A. In this case, we do not need to constrain the MWP-1A deglaciation history for BIIS, which significantly reduces the uncertainty for interpreting the non-local ice sheet contribution to the local MWP-1A rise magnitude across Scotland. As far as we know, this condition cannot be met anywhere outside Scotland because the other major ice sheets still existed during MWP-1A. In particular, the uncertainty associated with local ice sheet change will cause huge uncertainty for interpreting the near-field RSL data, and thus for deducing the meltwater signal from the NAIS, SIS and AIS. This large uncertainty will strongly hinder the differentiating power of that near-field site.

We do notice that there are two more near-field sites available in Norway (Vorren et al., 1988, Vasskog et al., 2019), which we believe have not been used for constraining MWP-1A sources. But as mentioned above, with large uncertainty associated with the SIS melting history during MWP-1A makes it difficult to use these sites to further refine the MWP-1A sources.

Intermediate-field sites that are relatively close to one of the major ice sheets, would be really useful for constraining relative contributions to MWP-1A, but there are often additional challenges to overcome when interpreting such data. We discuss one such site, on the Argentine Shelf, in Supplementary Section S2 (lines 788-791 in track change word file).

Vorren, T.O., Vorren, K.D., Alm, T., Gulliksen, S. and Løvlie, R., (1988) 'The last deglaciation (20,000 to 11,000 BP) on Andoya, Northern Norway'. *Boreas*, 17(1), pp.41-77.

Vasskog, K., Svendsen, J.I., Mangerud, J., Agasøster Haaga, K., Svean, A. and Lunnan, E.M., (2019) 'Evidence of early deglaciation (18 000 cal a bp) and a postglacial relative sea-level curve from southern Karmøy, south-west Norway'. *Journal of Quaternary Science*, 34(6), pp.410-423.

Although not at MWP1a, the recent Pico et al., Sci Adv, 2020 study used sea level at the Bering Strait to fingerprint melt sources during the Younger Dryas, and concluding that a large portion of the western North American ice sheets melted during this time. They argue that this result precludes the western North American ice sheets from contributing substantially to MWP-1a (which possibly contradicts the results in this study, depending on the geometry of ice melt needed to fit records in Scotland and Barbados).

We thank the reviewer for suggesting this interesting paper for comparison and now include reference to it in our discussion (lines 243-261 in track change word file). To compare our results with the ice history suggested by Pico et al., (2020), we have separated our NAIS MWP-1A melt geometry into contributions from the Western and Eastern Laurentide Ice Sheets (WLIS and ELIS), following the definition of separating these two ice sheets along the 110° W longitude in Pico et al., (2020).

The MWP-1A ice geometry used in our study (Lambeck et al., 2017) shows a ~58% North American contribution from the ELIS, consistent with the ~60% contribution represented in the ICE6G_C reconstruction used in Pico et al., (2020) and another recent ice sheet reconstruction (GLAC-1D from Tarasov et al., 2012; ~55%). Since Pico et al., did not modify the ELIS deglaciation history, our result agrees well with Pico et al., in terms of the ELIS contribution.

The question now is whether the WLIS melted during MWP-1A or during the Younger Dryas or both? Currently, there are four recent NAIS reconstructions available (Lambeck et al., 2017; Gowan et al., 2016; Peltier et al., 2015; Tarasov et al., 2012), and they all proposed that the WLIS along with the Cordilleran Ice Sheet should be a contributor to MWP-1A, which is consistent with the NAIS MWP-1A melt geometry used in this study. The study by Pico et al., (2020) did not focus on MWP-1A, but instead, they made an assumption of no WLIS contribution to MWP-1A in order to fit their Younger Dryas data, which require significant ice melt from the WLIS after MWP-1A. We are unable to identify any robust observational data to support this assumption and suggest that additional near-field glaciological evidence is required to resolve this issue, but restate that our results are consistent with all four regional reconstructions mentioned above.

Regarding the fit of the WLIS and ELIS fingerprints to Scotland and Barbados records and the ability of the RSL data to separate the contribution from the two ice sheet sectors, we have carried out some additional experiments, detailed in the response to the comment below.

Gowan, E. J., Tregoning, P., Purcell, A., Montillet, J. P., & McClusky, S. (2016) 'A model of the western Laurentide Ice Sheet, using observations of glacial isostatic adjustment' *Quaternary Science Reviews*, 139, 1-16.

Line 93: Wording is a bit confusing, I think it would make sense to start with something like the sentence on line 100 "We determine all three corrections...", because I felt confused on how the corrections were done, but then this was explained after.

We have modified this sentence (line 75 in track change word file) for better coherence.

Line 109: What was the requirement for sea level records used in this study? Is there a certain robustness for chronology across MWP-1a that each site must have?

The criterion for a sea-level site to be included in our analysis is that local RSL is constrained by high quality sea-level index points both before and after MWP-1A.

The requirements for an individual sea-level record to be used in this study are:

- 1) The sea-level data is publicly available and with robust relationship to sea level at the time of formation (indicative meaning).
- 2) Ideally, the mean age of that sea-level data needs to be within the defined conservative MWP-1A time window of 14.65-14.0 ka BP. But for some sites with insufficient temporal sea-level data coverage, we also include those sea-level data with 2σ age error bar extending into this time window.
- 3) We have excluded all sea-level data that are clearly stated as not *in situ* by original studies.
- 4) We have applied standard U-series age reliability filters (% calcite, $\delta^{234}\text{U}_{\text{initial}}$, [^{232}Th]).
- 5) For replicated ages, we use the inverse weighted mean value/distribution of each replicate group data.

All these five points are stated in the manuscript in lines 86-98 (track change word file) or in the **Method** *Monte Carlo linear regression* section.

I realize this is a fingerprinting study, but I wonder if the authors think it would be valuable to look at a longer timescale for RSL records (perhaps this would expand the number of records, which could include more near field sites) and consider the longer viscoelastic sea level response to the inferred ice melting scenario? I realize this is outside the scope of the present study, however I think this could potentially be highlighted as a future direction or it could be commented on why this elastic fingerprinting method is more appropriate.

This is an interesting point raised by the reviewer. We agree that including a longer timescale of RSL records and using them to constrain the viscoelastic sea-level response to the inferred ice melting scenario would be valuable. For example, longer records could be used to constrain the global ice volume before and after MWP-1A and therefore constrain the absolute magnitude of MWP-1A.

However, there are two challenges to carrying out longer timescale viscoelastic calculation. The first one is that a longer timescale calculation would require a continuous global deglaciation history, which is hard to constrain, especially for the period before MWP-1A. The second problem is that, for the viscoelastic response, the longer the time scale we consider the more important the Earth model parameters will become. Given that Earth rheology is poorly-constrained and spatially variable, this would become a big source of uncertainty.

Instead, our elastic fingerprinting method focuses on the centennial time scale of MWP-1A. This means that our results do not rely strongly on the ice history before MWP-1A or the unknown Earth rheology. We do quantify the viscous contribution to MWP-1A RSL change, and this enables us to invert the sources of MWP-1A in a linear manner using elastic fingerprinting, which is computationally very cheap compared to running a similar number of (~20,000) forward GIA models. We emphasise the importance of considering viscous effects during MWP-1A in our manuscript, but given the complications described above, we do not recommend that the method be used to investigate longer-term ice sheet and sea-level change.

Figure 2: It would be helpful to label HYD and NOG by the country they are in (or Great Barrier Reef) for easier reference. The other geographic locations are more intuitive.

These acronyms are used from Fig. 1 to Fig. 4, we suggest not to change the label in Fig. 2 in order to retain the consistency between these figures. Instead, we have added “both HYD and NOG are from the Great Barrier Reef” in the Fig.1 caption as a reminder for readers.

Figure 2: Why are there sometimes clouds of probability above the data points?

Many fossil corals are only identified to genus rather than the species level which often leads to multi-modal empirical depth distributions (i.e., the depth distributions are an amalgamation of many species' individual depth tolerances). For example, most of the samples from Tahiti during this interval are *Porites* sp. Modern specimens are mostly found in the upper water column (0-15 m) but a significant proportion are also found at deeper depths (40-50 m). When these modern depth distributions are included in our sea-level reconstructions, the deeper dwelling 'peak' results in the probability clouds above the data points. The bimodal living depth of this species is documented on lines 124-125 of our revised manuscript (track change word file).

Figure 3: Is the right side of the y axis meters of GMSL? If so this should be labeled. If not, I am not sure what this represents. I also think the “local MWP-1a magnitude” should perhaps be “local MWP-1a relative sea-level change magnitude”, if I am correct in understanding that the number represents the total change in RSL over MWP-1a at each site.

We thank the reviewer for pointing this out, we have amended this figure as suggested.

Figure 4: Since this figure does not compare RSL predictions using the different ice model modifications (ANU vs ANU-MWP), I'm not sure as is that it helps the reader understand why the best fit model is actually an improved fit. Perhaps the RSL predictions with the standard ANU model can be shown as a comparison? Or a misfit using the two RSL predictions?

We modified the ANU model and plot the revised RSL estimates to illustrate the validity of our inversion under a longer time scale viscoelastic calculation, but there is no specific intent to seek a better/optimum fit to the longer-term RSL data. The original ANU model produces a RSL oscillation at Arisaig, and this is removed when we adopt our MWP-1A solution. However, as the total MWP-1A GMSL rise magnitude is similar between the two models, the difference between original and revised RSL predictions for other far-field sites would be really small. In light of this comment, we have relabeled this as “Optimum Earth model for Scotland” (instead of “Best fit model”), and added the following clarification in the figure caption “Combining our MWP-1A solution with this optimum Earth model in Scotland, we achieve a good fit to RSL data and meanwhile avoid a local RSL oscillation.”.

For the Scottish site, it might be helpful to label on the figure “stratigraphy shows no oscillation” since this site requires a different interpretation from the others.

We have amended this figure as suggested.

Line 254: What about considering different parts of ice sheets? For example eastern vs western North American ice sheets? Geometry of ice melt is important for the elastic fingerprint, although these sites might be far away enough from ice sheets that it doesn't matter. Would be interesting to see sensitivity to ice sheet geometry.

We thank the reviewer for providing this interesting suggestion. We have carried out an experiment to treat the WLIS and ELIS separately based on their MWP-1A melt geometries from ICE6G_C. Using these two sea-level fingerprints together with sea-level fingerprints of the SIS and AIS, our inversion yields 4.4 m and 8.7 m contributions from the WLIS and ELIS, respectively, with 1.5 m and 3.5 m AIS and SIS contributions, similar to before (results shown in Supplementary Table S3). The most distinct difference between the WLIS and ELIS fingerprints is that the WLIS produces a significantly larger RSL rise in Barbados and Northwest Scotland (0.96 and 0.93) than the ELIS (0.69 and 0.64). Because the observed RSL rise in Barbados and Northwest Scotland is significantly lower than the global average during MWP-1A, a larger ELIS contribution is preferred. However, the Barbados and Scotland records cannot robustly constrain the WLIS and ELIS contributions, which are represented by large 95% uncertainty ranges of 0-12.5 m and 0-16.8 m for WLIS and ELIS, respectively. These results are reported in our revised manuscript.

We generated NAIS sea-level fingerprints using four NAIS reconstructions (Lambeck et al., 2017; Peltier et al., 2015; Tarasov et al., 2012; Gregoire et al., 2012, fingerprint of which was taken from Gomez et al., 2015), all of which resulted in very similar sea-level fingerprint values at our six sea-level sites, and negligible differences in our inversion results (Supplementary Table S3). We have added a Supplementary Table S3 to illustrate the uncertainty and sensitivity associated with different ice melt geometries. Some statements of these experiments have been added in lines 172-174 and 476-480 in track change word file.

Line 372: I would disagree with this sentence. The saddle-collapse mechanism has been modeled with dynamic ice sheet simulations forced with a deglacial climate, however the timing of saddle collapse in this modeling is not MWP-1a. Rather Gregoire et al. attribute a mechanism for a rapid melting of the North American ice sheets with MWP-1a. In model time, the timing is closer to Younger Dryas, which suggests that the relationship of an ice

saddle collapse with Bolling Allerod warming is not so clear.

We have modified our manuscript to “Rapid disintegration of the NAIS and SIS was proposed to be consistent with the operation of ice-sheet saddle collapse...” and use this sentence to lead into a discussion on the controversial NAIS and SIS ice dynamic behavior during MWP-1A. This ice dynamic uncertainty, in turn, leads into an ice melt geometry uncertainty analysis.

Indeed Pico et al. 2020 suggest that the North American saddle melted primarily during the Younger Dryas from 13-11.5 ka, rather than during MWP-1a, based on fingerprinting the flooding history of the Bering Strait. The study explores different MWP-1a melting scenarios in the supplement, and finds that most other combinations of ice sheet melt would fit the Bering Strait record. However, this result requires that ice melt during MWP-1a in North America is sourced from the eastern Laurentide, and not the ice sheet saddle, as suggested by Gregoire et al. This Pico et al. study should perhaps be brought into the discussion to frame the debate around potential meltwater sources for MWP-1a.

Pico, T., Mitrovica, J. X. and Mix, A. C. (2020) ‘Sea level fingerprinting of the Bering Strait flooding history detects the source of the Younger Dryas climate event’, *Science Advances*.

We have responded to this potential contradiction with Pico et al., 2020 in our responses above. And we have also incorporated the results by Pico et al., 2020 into our discussion between line 243 and 261 (in track change word file).

Consequently, I am curious whether the sensitivity of Barbados to North American ice sheets could differentiate between eastern and western sources. Would it be possible to test this by treating these two areas separately? It would be interesting to learn whether the data at Barbados requires ice melt from a certain region of the North American ice sheets, and whether the results in this paper are consistent or not with the conclusions drawn in Pico et al.

The results of our additional experiments are shown above. In general, our ELIS contribution agrees with the ICE6G_C model. However, our results do require a 30-40% contribution from the WLIS, which is inconsistent with conclusions drawn in Pico et al., 2020. Indeed, the conclusions by Pico et al., relied on a 5.7 m Antarctic contribution to MWP-1A, which would be largely implausible based on our inversion results. Several recent studies by Gomez et al., 2020, Albrecht et al., 2020 also suggest the Antarctic contribution to MWP-1A should be <2 m. We therefore suggest the WLIS should at least make some contribution to MWP-1A instead of the 0 m suggested by Pico et al.

Gomez, N., Weber, M. E., Clark, P. U., Mitrovica, J. X., & Han, H. K. (2020) ‘Antarctic ice dynamics amplified by Northern Hemisphere sea-level forcing’ *Nature*, 587(7835), 600-604.

Albrecht, T., Winkelmann, R. & Levermann, A. Glacialcycle simulations of the Antarctic Ice Sheet with the Parallel Ice Sheet Model (PISM)–Part 2: Parameter ensemble analysis. *Cryosphere* 14 (2020).

Figure S5: I think that the ice melt geometries are very important, and should at least be described in the main text. After reading the main text, I was unsure if the ice melting scenarios were uniform ice thicknesses or not. These figures are helpful, but if it is not possible to include them, it would be helpful to describe in some way that the ice histories

used are characterized by spatially heterogeneous melting, and perhaps even explaining what regions contain the most ice melt within these different ice sheets.

We agree with the reviewer that this figure is very important and we have combined this figure with Figure 1 (i.e., the sea-level fingerprint figure) in the main text.

REVIEWERS' COMMENTS

Reviewer #1 (Remarks to the Author):

My comments have been fully addressed by the authors and I recommend this excellent manuscript to be published.

Reading through the manuscript again I spotted some minor things regarding wording and formatting:

Line 10: "Ongoing uncertainty regarding the sources of meltwater limits..." I suggest that this is rephrased to something like: "Considerable uncertainty regarding the sources of meltwater limits ..."

Line 118: «asPoritessp.1.» please correct

Line 119: «0-15 m and 40-50m40.» Should this be a reference to Ref. 40?

Reviewer #3 (Remarks to the Author):

Thank you to Lin and coauthors for the clear, thorough, and thoughtful response to reviewer comments. I have read through this document and the updated manuscript. All the issues I raised have been addressed. My only remaining suggestion is to include a reference to the Great Barrier Reef in all figures that include HYD and NOG to reduce the need for readers to switch between figures to interpret the results. Even without including this suggestion, I would happily

REVIEWER COMMENTS

Reviewer #1 (Remarks to the Author):

My comments have been fully addressed by the authors and I recommend this excellent manuscript to be published.

We thank the reviewer for this positive feedback on our revisions. The format problems below are mainly caused by the Latex-Word transition, which we failed to spot. We thank the reviewer for noticing them.

Reading through the manuscript again I spotted some minor things regarding wording and formatting:

Line 10: “Ongoing uncertainty regarding the sources of meltwater limits...” I suggest that this is rephrased to something like: “Considerable uncertainty regarding the sources of meltwater limits ...”

We thank the reviewer for this nice suggestion, we have amended this as suggested.

Line 118: «asPoritessp.1.» please correct

We have amended this problem.

Line 119: «0-15 m and 40-50m40.» Should this be a reference to Ref. 40?

Yes, 40 is the reference number, we have fixed this format problem.

Reviewer #3 (Remarks to the Author):

Thank you to Lin and coauthors for the clear, thorough, and thoughtful response to reviewer comments. I have read through this document and the updated manuscript. All the issues I raised have been addressed. My only remaining suggestion is to include a reference to the Great Barrier Reef in all figures that include HYD and NOG to reduce the need for readers to switch between figures to interpret the results. Even without including this suggestion, I would happily recommend this manuscript for publication as is. Congratulations on an excellent paper!

We thank the reviewer for this positive feedback on our revisions. We agree that including a reference to the Great Barrier Reef in our figures would be beneficial to the readers. We have amended all the figures accordingly.